# MODEL ALREADY KNOWS THE BEST NOISE: BAYESIAN ACTIVE NOISE SELECTION VIA ATTENTION IN VIDEO DIFFUSION MODEL

**Kwanyoung Kim**[1,†]**, Sanghyun Kim**[2]**,**
Department of AI Convergence, Gwangju Institute of Science and Technology (GIST) [1],
Samsung Research[2]
k_0.kim@gist.ac.kr, sanghn.kim@samsung.com

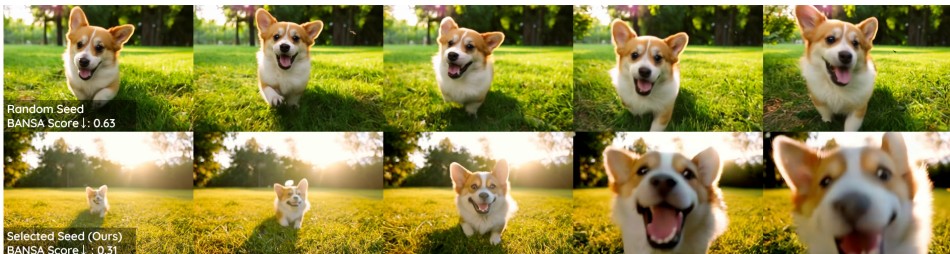

"A joyful Corgi with a fluffy coat and perky ears frolics in a sunlit park, the golden hues of sunset casting a warm glow on the scene. The camera zooms in on the Corgi's expressive face, capturing its bright eyes and wide, happy grin...."

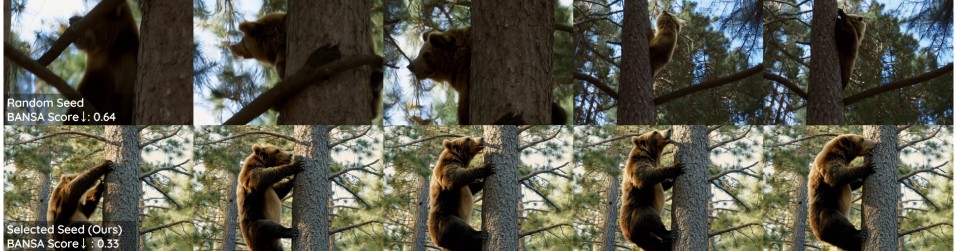

"A majestic brown bear, with its thick fur glistening in the dappled sunlight, begins its ascent up a towering pine tree in a dense forest. The bear's powerful claws grip the rough bark as it climbs higher, its muscles rippling with each movement..."

Figure 1: **Random Seed vs. Ours.** We propose ANSE, a noise selection framework, and the BANSA Score, an uncertainty-based metric. By selecting initial noise seeds with lower BANSA scores, which indicate more certain noise samples, ANSE improves video generation performance.

## ABSTRACT

The choice of initial noise strongly affects quality and prompt alignment in video diffusion; different seeds for the same prompt can yield drastically different results. While recent methods use externally designed priors (e.g., frequency filtering or inter-frame smoothing), they often overlook internal model signals that indicate inherently preferable seeds. To address this, we propose **ANSE** (**A**ctive **N**oise **S**election for **G**eneration), a model-aware framework that selects high-quality seeds by quantifying attention-based uncertainty. At its core is **BANSA** (Bayesian Active Noise Selection via Attention), an acquisition function that measures entropy disagreement across multiple stochastic attention samples to estimate model confidence and consistency. For efficient inference-time deployment, we introduce a Bernoulli-masked approximation of BANSA that estimates scores from a single diffusion step and a subset of informative attention layers. Experiments across diverse text-to-video backbones demonstrate improved video quality and temporal coherence with marginal inference overhead, providing a principled and generalizable approach to noise selection in video diffusion. See our project page: https://anse-project.github.io/anse-project/.

[†]First and corresponding author, this work was done while the author was at Samsung Research.

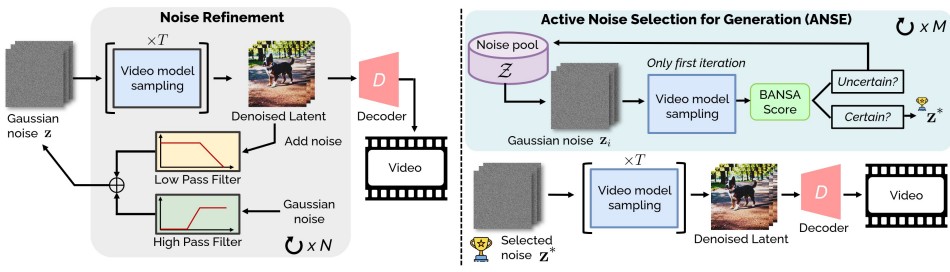

(a) Conventional Noise Initialization approaches     (b) Our proposed framework, ANSE

Figure 2: **Conceptual comparison of noise initialization.** **(a)** Prior methods (Wu et al., 2024; Yuan et al., 2025) refine noise via frequency priors and full diffusion sampling, leading to high cost. **(b)** Our method instead selects noise seeds by estimating attention-based uncertainty at the first denoising step, enabling efficient, model-aware selection.

# 1 INTRODUCTION

Diffusion models have rapidly established themselves as a powerful class of generative models, demonstrating state-of-the-art performance across images and videos (Rombach et al., 2022; Esser et al., 2024; Xie et al., 2024; Chen et al., 2024b;a; Wang et al., 2025; Yang et al., 2024; Zheng et al., 2024; Team, 2025). In particular, Text-to-Video (T2V) diffusion models have received increasing attention for their ability to generate temporally coherent and visually rich video sequences. To achieve this, most T2V model architectures extend Text-to-Image (T2I) diffusion backbones by incorporating temporal modules or motion-aware attention layers (Blattmann et al., 2023; He et al., 2022; Wang et al., 2023; Chen et al., 2023; 2024a; Guo et al., 2024b; Wang et al., 2025). Furthermore, other works explore video generative structures, such as causal autoencoders or video autoencoder-based models, which aim to generate full video volumes rather than a sequence of independent frames (Hong et al., 2022; Yang et al., 2024; HaCohen et al., 2024; Kong et al., 2024; Zheng et al., 2024; Team, 2025).

Beyond architectural design, another promising direction lies in improving noise initialization at inference time for T2I and T2V generation (Guo et al., 2024a; Eyring et al., 2024; Chen et al., 2024c; Xu et al., 2025). This aligns with the growing trend of inference-time scaling, observed not only in Large Language Models (Brown et al., 2024; Snell et al., 2024) but also in diffusion-based generation systems (Ma et al., 2025). Due to the iterative nature of the diffusion process, the choice of initial noise profoundly influences video quality, temporal consistency, and prompt alignment (Ge et al., 2023; Wu et al., 2024; Qiu et al.; Yuan et al., 2025). As illustrated in Figure 1, the same prompt can lead to drastically different videos depending solely on the noise seed, motivating the need for intelligent noise selection.

Recent approaches tackle this by introducing external noise priors. PYoCo (Ge et al., 2023) enforces inter-frame dependent patterns for coherence but requires heavy fine-tuning. FreeNoise (Qiu et al.) reschedules noise across time with a fusion strategy, FreeInit (Wu et al., 2024) preserves low-frequency components via frequency filtering, and FreqPrior (Yuan et al., 2025) extends this with Gaussian-shaped priors and partial sampling. While effective, these methods rely on external priors and repeated full diffusion passes, while ignoring internal model signals that identify inherently preferable seeds.

To address this limitation, we propose a model-aware noise selection framework, **ANSE** (**A**ctive **N**oise **S**election for **Ge**neration), grounded in Bayesian uncertainty. At the core of ANSE is **BANSA** (**B**ayesian **A**ctive **N**oise **S**election via **A**ttention), an acquisition function that identifies noise seeds inducing confident and consistent attention behaviors under stochastic perturbations. A conceptual comparison between our method and prior frequency-based approaches is illustrated in Figure 2, highlighting the difference between external priors and model-informed uncertainty estimates.

Unlike BALD (Gal et al., 2017), which estimates uncertainty from classification logits, BANSA measures entropy in attention maps, arguably the most informative signals in diffusion. It compares the mean of per-pass entropies with the entropy of the mean map, capturing both uncertainty and cross-pass disagreement. A lower BANSA score indicates more confident, consistent attention and empirically correlates with more coherent video generation (Fig. 1). To make BANSA inference-

friendly, we approximate it with Bernoulli-masked attention, yielding multiple stochastic attention samples from a single forward pass. We further cut cost by evaluating early denoising steps and only a subset of informative layers selected via correlation analysis. Our contributions are threefold:

- We present **ANSE**, the first active noise selection framework for video diffusion, grounded in a Bayesian formulation of attention-based uncertainty.

- We introduce **BANSA**, an acquisition function that measures attention consistency under stochastic perturbations, enabling model-aware noise selection without retraining.

- Our method improves video quality and temporal consistency across diverse text-to-video backbones, with only marginal inference overhead.

## 2  PRELIMINARY

**Video Diffusion Models**  Diffusion models (Ho et al., 2020; Song et al., 2021b) have achieved strong results across generative tasks. In T2V, pixel-space diffusion is costly, so most video diffusion models adopt latent diffusion and operate in a compressed latent space. A video autoencoder with encoder $\mathcal{E}$ and decoder $\mathcal{D}$ reconstructs $\mathbf{x}$ as $\mathbf{x} = \mathcal{D}(\mathcal{E}(\mathbf{x}))$. Let $\mathbf{z}_0 = \mathcal{E}(\mathbf{x})$. The forward diffusion then progressively adds noise over time:

$$\mathbf{z}_t = \sqrt{\bar{\alpha}_t}\mathbf{z}_0 + \sqrt{1 - \bar{\alpha}_t}\,\boldsymbol{\epsilon}, \quad \boldsymbol{\epsilon} \sim \mathcal{N}(0, \mathbf{I}), \quad t = 1, \dots, T,$$

where $\bar{\alpha}_t$ is a pre-defined variance schedule. To learn the reverse process, a denoising network $\boldsymbol{\epsilon}_\theta$ is trained using the denoising score matching loss (Vincent, 2011):

$$\mathcal{L}_\theta = \mathbb{E}_{\mathbf{z}_t, \boldsymbol{\epsilon}, t} \left[ \|\boldsymbol{\epsilon}_\theta(\mathbf{z}_t, \mathbf{c}, t) - \boldsymbol{\epsilon}\|^2 \right],$$

where $\mathbf{c}$ is the conditioning text. Sampling starts from Gaussian noise $\mathbf{z}_T \sim \mathcal{N}(0, \mathbf{I})$ and uses a deterministic DDIM solver (Song et al., 2021a). Each step updates as:

$$\mathbf{z}_{t-1} = \sqrt{\bar{\alpha}_t}\,\hat{\mathbf{z}}_0(t) + \sqrt{1 - \bar{\alpha}_{t-1}}\,\boldsymbol{\epsilon}_\theta(\mathbf{z}_t, \mathbf{c}, t),$$

where the denoised latent estimate $\hat{\mathbf{z}}_0(t) := \frac{\mathbf{z}_t - \sqrt{1 - \bar{\alpha}_t}\,\boldsymbol{\epsilon}_\theta(\mathbf{z}_t, \mathbf{c}, t)}{\sqrt{\bar{\alpha}_t}}$ is obtained using Tweedie's formula (Efron, 2011; Kim & Ye, 2021). This iterative process continues until $t = 1$, yielding the final denoised latent $\mathbf{z}_0$, which is decoded into a video via $\mathcal{D}$.

**Bayesian Active Learning by Disagreement (BALD)**  Active learning selects the most informative samples from an unlabeled pool to improve training. Acquisition functions are commonly uncertainty-based (Houlsby et al., 2011; Gal et al., 2017; Kirsch et al., 2019; Yoo & Kweon, 2019) or distribution-based (Mac Aodha et al., 2014; Yang et al., 2015; Sener & Savarese, 2017; Sinha et al., 2019), and some use external modules such as auxiliary predictors (Yoo & Kweon, 2019; Tran et al., 2019; Kim et al., 2021). While most prior work targets image classification, we adapt uncertainty-based acquisition to text-to-video generation without additional models.

Predictive entropy is widely used, but it reflects data noise and does not isolate parameter uncertainty. BALD instead measures *epistemic* uncertainty via the mutual information between predictions $\mathbf{y}$ and parameters $\theta$:

$$\text{BALD}(\mathbf{x}) = \mathcal{H}[p(\mathbf{y}|\mathbf{x})] - \mathbb{E}_{p(\theta|\mathcal{D}_U)}\left[\mathcal{H}[p(\mathbf{y}|\mathbf{x}, \theta)]\right], \tag{1}$$

where $\mathcal{H}[p] = -\sum_y p(y) \log p(y)$ is Shannon entropy (Shannon, 1948). The first term is the entropy of the mean prediction, and the second is the average entropy across stochastic forward passes. A high BALD score means predictions are confident yet disagree, indicating high epistemic uncertainty. Since the posterior over $\theta$ is intractable, BALD is approximated using $K$ stochastic forward passes (e.g., Monte Carlo dropout):

$$\widehat{\text{BALD}}(\mathbf{x}) = \mathcal{H}\left[\frac{1}{K}\sum_{k=1}^{K} p^{(k)}(\mathbf{y}|\mathbf{x})\right] - \frac{1}{K}\sum_{k=1}^{K} \mathcal{H}\left[p^{(k)}(\mathbf{y}|\mathbf{x})\right]. \tag{2}$$

We reinterpret BALD for inference-time generative modeling. Rather than selecting samples for labeling, we apply BALD to rank noise seeds by their epistemic uncertainty. Selecting seeds with lower BALD scores results in more stable model behavior and leads to higher-quality generations.

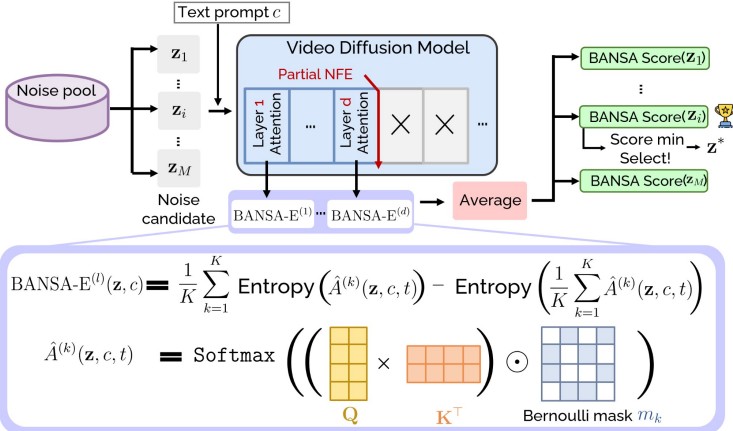

Figure 3: **Overview of our BANSA-based noise selection process.** Given a text prompt $c$, we compute BANSA scores for multiple noise seeds $\{\mathbf{z}_1, \ldots, \mathbf{z}_M\}$ using Bernoulli-masked attention maps from selected layers at an early diffusion step. The seed with the lowest score, indicating confident and consistent attention, is selected for generation.

## 3 METHODS

We propose **ANSE**, a framework for selecting high-quality noise seeds in T2V diffusion based on model uncertainty (Fig. 2). ANSE centers on **BANSA**, an acquisition function that transfers uncertainty criteria from classification to the attention space of generative diffusion models (Sec. 3.1). For efficient inference, we approximate BANSA via Bernoulli-masked attention sampling (Sec. 3.2). To avoid redundancy, we select a representative attention layer using correlation-based linear probing (Sec. 3.3). The full pipeline appears in Fig. 3.

### 3.1 BANSA: BAYESIAN ACTIVE NOISE SELECTION VIA ATTENTION

We introduce **BANSA**, an acquisition function for selecting optimal noise seeds in T2V diffusion. Unlike classifiers with explicit predictive distributions, diffusion models do not expose such outputs, so we estimate uncertainty in the attention space where text and visual tokens align during generation. We treat attention maps as stochastic predictions conditioned on the seed $\mathbf{z}$, prompt $\mathbf{c}$, and timestep $t$. BANSA measures both disagreement and confidence across multiple attention samples, providing a BALD-style uncertainty criterion tailored to the generative setting.

**Definition 1** (BANSA Score). *Let $\mathbf{z}$ be a noise seed, $c$ a text prompt, and $t$ a diffusion timestep. Let $\mathbf{Q}(\mathbf{z}, c, t), \mathbf{K}(\mathbf{z}, c, t) \in \mathbb{R}^{N \times d}$ denote the query and key matrices from a denoising network $\epsilon_\theta$. The attention map is computed as:*

$$A(\mathbf{z}, c, t) := \texttt{Softmax}\left(\mathbf{Q}(\mathbf{z}, c, t)\,\mathbf{K}(\mathbf{z}, c, t)^\top\right) \in \mathbb{R}^{N \times N}. \tag{3}$$

*Let $\mathcal{A}(\mathbf{z}, c, t) = \{A^{(1)}, \ldots, A^{(K)}\}$ denote a set of $K$ stochastic attention maps obtained via forward passes with random perturbations (e.g., Bernoulli masking). The **BANSA score** is defined as:*

$$BANSA(\mathbf{z}, c, t) := \mathcal{H}\left(\frac{1}{K}\sum_{k=1}^{K} A^{(k)}\right) - \frac{1}{K}\sum_{k=1}^{K}\mathcal{H}(A^{(k)}), \tag{4}$$

where $\mathcal{H}(A) := \frac{1}{N}\sum_{i=1}^{N}\sum_{j=1}^{N} -A_{ij}\log A_{ij}$. This formulation captures both the sharpness (confidence) and the consistency (agreement) of attention behavior. BANSA can be applied to various attention types (e.g., cross-, self-, or temporal) and allows layer-wise interpretability.

Given a noise pool $\mathcal{Z} = \{\mathbf{z}_1, \ldots, \mathbf{z}_M\}$, we select the optimal noise seed that minimizes score:

$$\mathbf{z}^* := \arg\min_{\mathbf{z} \in \mathcal{Z}} \text{BANSA}(\mathbf{z}, c, t). \tag{5}$$

A desirable property of BANSA is that its score becomes zero when all attention samples are identical, reflecting complete agreement and certainty. We formalize this as follows:

---

**Algorithm 1:** Active Noise Selection with BANSA Score for Video Generation

---

**Input:** Text prompt $c$, noise pool $\mathcal{Z} = \{\mathbf{z}_1, \ldots, \mathbf{z}_M\}$, timestep $t$, cutoff layer $d^*$

1 **foreach** $\mathbf{z}_i \in \mathcal{Z}$ **do**

2  $\quad$ Compute BANSA score: $\widehat{\text{BANSA-E}}_{\leq d^*}(\mathbf{z}_i, c, t)$ via Eq. (7);

3 Select optimal noise: $\mathbf{z}^* = \arg\min_{\mathbf{z}_i} \widehat{\text{BANSA-E}}_{\leq d^*}(\mathbf{z}_i, c, t)$;

4 **return** *Generate video:* $\hat{v} = \texttt{SampleVideo}(\mathbf{z}^*, c, t)$;

---

**Proposition 1** (BANSA Zero Condition). *Let $\mathcal{A}(\mathbf{z}, c, t) = \{A^{(1)}, \ldots, A^{(K)}\}$ be a set of row-stochastic attention maps. Then:*

$$BANSA(\mathbf{z}, c, t) = 0 \quad \Leftrightarrow \quad A^{(1)} = \cdots = A^{(K)}.$$

The proof is deferred to the Appendix B. This result implies that minimizing the BANSA score encourages attention that is both confident and consistent under stochastic perturbations. Empirically, low BANSA correlates with better prompt alignment, temporal coherence, and visual fidelity. Thus, BANSA provides a principled, model-aware criterion for noise selection in T2V.

### 3.2 Stochastic Approximation of BANSA via Bernoulli-Masked Attention

While BANSA is principled for noise selection, computing it with $K$ independent passes per seed $\mathbf{z}$ is costly. We instead draw $K$ stochastic attention samples from a *single* pass via Bernoulli-masked attention. Rather than running $K$ passes, we inject stochasticity directly into the attention scores. For each $k = 1, \ldots, K$, sample a binary mask $m_k \in \{0, 1\}^{N \times N}$ i.i.d. from $\text{Bernoulli}(p)$ and compute masked attention map, $\hat{A}^{(k)}(\mathbf{z}, c, t) := A(\mathbf{z}, c, t) \odot m_k$, $\odot$ denotes element-wise multiplication and rows are re-normalized. Using these $K$ such samples, we define the approximate BANSA:

$$\text{BANSA-E}(\mathbf{z}, c, t) := \mathcal{H}\left(\frac{1}{K}\sum_{k=1}^{K} \hat{A}^{(k)}(\mathbf{z}, c, t)\right) - \frac{1}{K}\sum_{k=1}^{K}\mathcal{H}(\hat{A}^{(k)}(\mathbf{z}, c, t)). \tag{6}$$

By concavity of entropy, BANSA-E $\geq 0$. Although approximate, it efficiently captures attention-level epistemic uncertainty and serves as a practical surrogate for model-aware noise selection. [†] As shown in Table 4, experimental validation confirms that this method is sufficient for selecting optimal noisy samples from the model's perspective.

### 3.3 Layer Selection via Cumulative BANSA Correlation

BANSA can be computed at any layer, but behavior differs across depth. Using all layers is costly, so we truncate at the smallest depth $d^*$ where the average BANSA up to $d$ remains highly correlated with the full-layer score. Given a noise seed $\mathbf{z}_i \in \mathcal{Z} = \{\mathbf{z}_1, \ldots, \mathbf{z}_M\}$ and $L$ attention layers, we compute per-layer scores BANSA-E$^{(l)}(\mathbf{z}_i, c, t)$ and define the cumulative average up to layer $d$ as:

$$\widehat{\text{BANSA-E}}_{\leq d}(\mathbf{z}_i, c, t) := \frac{1}{d}\sum_{l=1}^{d} \text{BANSA-E}^{(l)}(\mathbf{z}_i, c, t). \tag{7}$$

To find $d^*$, we compute the Pearson correlation (Pearson, 1895) between the partial average $\widehat{\text{BANSA-E}}_{\leq d}$ and the full-layer average $\widehat{\text{BANSA-E}}_{\leq L}$, and choose the smallest $d$ such that $\text{Corr}\left(\widehat{\text{BANSA-E}}_{\leq d}, \widehat{\text{BANSA-E}}_{\leq L}\right) \geq \tau$. We validate this procedure using 100 prompts and 10 noise seeds across all evaluated models as detailed in Appendix D. As shown in Figure 7, the correlation quickly stabilizes at a moderate depth, allowing us to select $d^*$ without using all layers. We then define the BANSA score as $\widehat{\text{BANSA-E}}_{\leq d^*}$ to guide noise selection, as summarized in Algorithm 1. With $d^*$ fixed per model, it adds no runtime cost, and $\widehat{\text{BANSA-E}}_{\leq d^*}$ (truncated layer) consistently matches the full-layer score and robustly preserves generation quality across models, confirming its reliability as a practical surrogate as shown Appendix D and E (see detail).

---

[†]We use "Bayesian" in the sense of epistemic uncertainty estimation, following BALD (Houlsby et al., 2011), rather than a full posterior derivation.

## 4    ANALYSIS OF BANSA-SELECTED NOISE

In this section, we aim to investigate the property of noise seeds selected by BANSA, and consist of three experiments which declare our framework and interpret the noise seeds. Our framework does not seek a universal "golden" seed, but selects the most suitable one for each prompt by minimizing uncertainty. The effectiveness of a noise seed is therefore *prompt-dependent*: a seed that improves one prompt may degrade another.

**Experiment 1: Cross-Prompt Behavior.** To demonstrate this prompt-dependent behavior, We select a high-BANSA (worst-performing) seed from prompt A and applied it to prompts B and C (Table 1) and measures Subject and Background Consistency. The seed degraded quality for prompt C but improved performance for prompt B, confirming that noise effectiveness depends on prompt context rather than intrinsic seed quality.

Table 1: Cross-prompt evaluation of a high-BANSA seed from A.

| From | To | Subject. | BackGround |
|------|-----|----------|------------|
| A | B | 0.8491→0.8703 | 0.9157→0.9538 |
| A | C | 0.7683→0.7213 | 0.9368→0.9169 |

While Experiment 1 shows how noise effectiveness varies across prompts, the following two experiments examine whether similar patterns also emerge within a *fixed* prompt.

**Experiment 2: Intra-Prompt Attention Consistency.** To examine whether BANSA uncertainty relates to attention stability, we generate five samples per prompt (total 100 prompts), selected the highest- and lowest-scoring seeds, and computed pairwise Euclidean distances between their attention maps (Table 2). Low-BANSA seeds showed smaller intra-group distances, indicating more stable and consistent patterns, while high-BANSA seeds exhibited higher variability.

Table 2: Pairwise distances within attention maps.

| Type | High (intra) | Low (intra) | Cross |
|------|--------------|-------------|-------|
| Euclid. | 1.635 | 1.567 | 1.735 |

**Experiment 3: Latent Trajectory and Expressiveness.** Beyond attention maps, we further investigate latent-space dynamics. Across 100 prompts with five repetitions each, we compare high- and low-BANSA seeds over all denoising steps (Table 3). We measured two metrics: (1) Latent Trajectory Variation, obtained by applying a Butterworth low-pass filter to isolate low-frequency structural components (Wu et al., 2024; Yuan et al., 2025) and computing their temporal variation, where lower values indicate smoother and more stable trajectories; and (2) Intra-frame Variance, the average spatial variance within frames, where higher values indicate richer dynamic expressiveness. Low-BANSA seeds showed both lower trajectory variation and higher intra-frame variance, combining temporal stability with more expressive video generation.

Table 3: Latent stability and visual dynamics.

| Group | Traj. Var. ↓ | Var. ↑ |
|-------|--------------|--------|
| High BANSA | 52.34 | 0.041 |
| Low BANSA | 51.07 | 0.053 |

**Takeaway and Relation to Prior Work.** The three findings indicate that BANSA identifies seeds with lower uncertainty, leading to (1) prompt-specific improvements, (2) more stable attention, and (3) smoother yet more expressive latent dynamics. This is consistent with FreeInit Wu et al. (2024) and FreeQPrior (Yuan et al., 2025), which emphasize stable latent initialization and low-frequency structure. BANSA complements these insights by offering an uncertainty-driven selector that explains *why* such seeds generalize within a prompt while remaining prompt-dependent across prompts.

## 5    EXPERIMENTS

**Experimental Setting.** We evaluate **ANSE** on diverse T2V diffusion backbones—AnimateDiff (Guo et al., 2024b), CogVideoX-2B/5B (Yang et al., 2024), Wan2.1 (Team, 2025), and Hunyuan-Video (Kong et al., 2024). To rigorously evaluate our method, we follow each model's official sampling protocol; for resolution, Wan2.1 is evaluated at 480p and HunyuanVideo at 360p due to resource limits. Results for noise-prior refinement baselines (FreqPrior (Yuan et al., 2025)) are reported only on AnimateDiff, as they lack official support on the others and incur $\sim 3\times$ inference time. ANSE is orthogonal and can be combined with these priors. Unless noted, we use a noise pool of $M{=}10$ with $K{=}10$ stochastic passes per seed, and Bernoulli-masked attention with $p{=}0.2$. All experiments run on NVIDIA H100 GPUs. Further details are in the Appendix C.

Table 4: **Quantitative results on VBench using AnimateDiff, CogVideoX-2B and -5B.**

| Backbone Model | Method | Quality Score | Semantic Score | Total Score | Inference Time |
|---|---|---|---|---|---|
| AnimateDiff (Guo et al., 2024b) | Vanilla | 80.22 | 69.03 | 77.98 | 28.23s |
| | + Ours | **81.66** | **71.09** | **79.33** | 31.33s$_{(+10.98\%)}$ |
| | Freqprior | 81.22 | 70.45 | 79.07 | 58.01s$_{(+105.36\%)}$ |
| | + Ours | **82.23** | **73.23** | **80.43** | 61.12s$_{(+5.36\%)}$ |
| CogVideoX-2B (Yang et al., 2024) | Vanilla | 82.08 | 76.83 | 81.03 | 247.8s |
| | + Ours | **82.56** | **78.06** | **81.66** | 269.3s$_{(+8.67\%)}$ |
| CogVideoX-5B (Yang et al., 2024) | Vanilla | 82.53 | 77.50 | 81.52 | 667.3s |
| | + Ours | **82.70** | **78.10** | **81.71** | 754.1s$_{(+13.1\%)}$ |

Table 5: **Quantitative results on quality score metrics for HunyuanVideo and Wan2.1**

| Backbone Model | Method | Subject Consistency ↑ | Background Consistency ↑ | Motion Smoothness ↑ | Aesthetic Quality ↑ | Imaging Quality ↑ | Temporal Flickering ↑ | Inference Time ↓ |
|---|---|---|---|---|---|---|---|---|
| HunyuanVideo (Kong et al., 2024) | Vanilla | 0.9562 | 0.9656 | 0.9850 | **0.6276** | 0.6137 | 0.9921 | 52.3s |
| | + Ours | **0.9612** | **0.9661** | **0.9858** | 0.6268 | **0.6151** | **0.9938** | 59.8s$_{(+14.34\%)}$ |
| Wan2.1 (Team, 2025) | Vanilla | 0.9562 | 0.9656 | 0.9850 | **0.6276** | 0.6137 | 0.9943 | 43.8s |
| | + Ours | **0.9612** | **0.9661** | **0.9858** | 0.6268 | **0.6151** | **0.9956** | 50.7s$_{(+15.75\%)}$ |

**Evaluation Metric.** We evaluate with VBench (Huang et al., 2024), which reports a quality score and a semantic score, combined into a total score on a 0–100 scale. For AnimateDiff and CogVideoX-2B/5B, we report both quality and semantic scores. For HunyuanVideo and Wan2.1, we restrict evaluation to the six quality dimensions due to computational constraints, while noting that semantic evaluation can also be applied in principle. All reported results are obtained by repeating each evaluation dimension five times and averaging the scores.

**Quantitative Comparison.** As shown in Table 4, ANSE consistently improves performance across diverse T2V backbones. On AnimateDiff, ANSE outperforms the vanilla baseline and further demonstrates clear advantages over noise-initialization methods such as FreqPrior (Yuan et al., 2025).

Notably, ANSE is also fully compatible with FreqPrior, achieving even higher scores when combined. Table 4 also reports results on CogVideoX-2B and -5B, which adopt the more advanced MMDiT architecture. ANSE improves quality, semantic, and total VBench scores on both scales, demonstrating robust generalization across architectures and model sizes.

Table 5 presents additional results on recent state-of-the-art models, HunyuanVideo and Wan2.1. ANSE achieves consistent improvements across quality metrics, underscoring its plug-and-play applicability to a wide spectrum of video diffusion models—from U-Net to MMDiT, and from mid-scale to the latest state-of-the-art systems. The statistical signicanse analysis of these are provied in Appendix J.

Table 6: **Comparison with FVMD metrics on MSRVTT dataset.**

| Backbone Model | Methods | FVMD ↓ |
|---|---|---|
| HunyuanVideo (Kong et al., 2024) | Vanilla | 17641.19 |
| | + Ours | **16491.68** |
| Wan2.1(1.3B) (Team, 2025) | Vanilla | 16495.59 |
| | + Ours | **14306.19** |

We further evaluate motion quality using the Fréchet Video Motion Distance (FVMD) (Liu et al., 2024) on MSR-VTT (Xu et al., 2016), as shown in Table 6. We sample 200 text prompts from the test split and generate videos with ANSE, using the corresponding real videos as the reference distribution. ANSE consistently obtains lower FVMD scores, indicating improved motion fidelity and reinforcing the motion-related gains observed in VBench.

**Qualitative Comparison.** Figure 4 presents representative qualitative comparisons on CogVideoX-2B, CogVideoX-5B, and Wan2.1 with and without ANSE. Our approach consistently enhances semantic fidelity, motion realism, and visual clarity across diverse prompts. For example, in separate prompts such as *"exploding"* and *"descends gracefully"*, ANSE captures critical semantic transitions—generating visible explosions in the former and preserving smooth temporal continuity in the latter. In other prompts like *"koala playing the piano"* and *"tasting beer"*, it produces anatomically coherent bodies with natural, expressive motion. These examples highlight ANSE's ability to improve spatial-temporal fidelity while generalizing effectively to large-scale video diffusion models. Additional qualitative results, including other backbones, are provided in Appendix L.

**Computational Cost.** As shown in Table 4 and Table 5, ANSE introduces only a minimal increase in inference time while delivering consistent quality gains. On AnimateDiff, inference time increases by just +10.98%, compared to more than +105% when combined with FreqPrior and over +200%

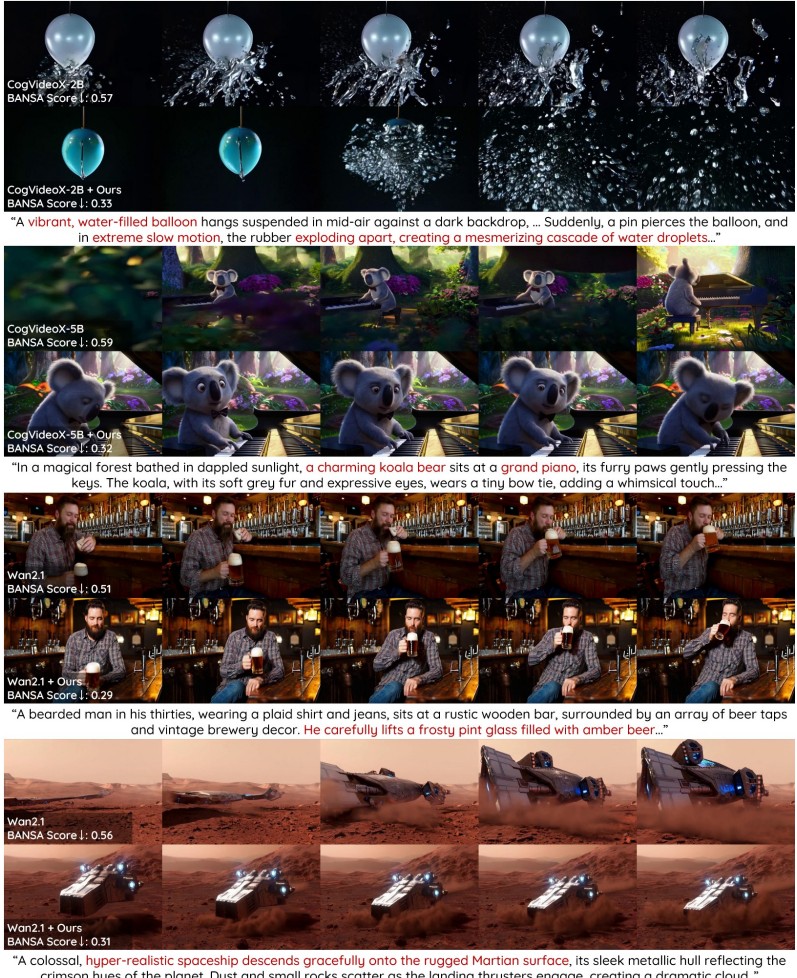

Figure 4: **Qualitative comparison with and without ANSE.** (CogvideoX-2B, 5B and Wan2.1). With ANSE, videos exhibit improved visual quality, better text alignment, and smoother motion transitions compared to the baseline. White numbers show BANSA scores for the same prompt.

for FreeInit. On CogVideoX models, the overhead is similarly small: +8% on the 2B variant and +13% on the 5B variant. For more recent architectures, ANSE adds only +14% on HunyuanVideo and +15% on Wan2.1. The overhead comes only from seed evaluation and leaves the sampling process and memory usage unchanged. In contrast, FreeInit and FreqPrior require multiple full passes, causing large slowdowns. ANSE improves quality across diverse backbones while keeping inference overhead below +15%, making it a practical plug-and-play solution for video diffusion.

## 6 ABLATION STUDY

**Comparison of Acquisition Functions.** Using CogVideoX-2B, we compare BANSA with random sampling, entropy-based selection, and two variants: BANSA (B) with Bernoulli masking and BANSA (D) with dropout-based stochasticity (Table 7). All methods improve over the baseline, but BANSA (B) consistently achieves the best scores across quality, semantic, and total metrics. This suggests that Bernoulli masking better captures attention-level uncertainty than dropout, underscoring the importance of modeling uncertainty in line with the model's structure.

**Effect of Ensemble Size $K$.** We investigate how the number of stochastic forward passes $K$ influences subject and background consistency (Table 8). Both metrics improve as $K$ increases from 1 to 10, indicating that larger ensembles provide more reliable noise evaluation. Performance saturates at $K = 10$, which we set as the default in all experiments.

Table 7: **Comparison of different acquisition functions for noise selection.**

| Method | Quality Score | Semantic Score | Total Score |
|---|---|---|---|
| Random | 82.08 | 76.83 | 81.03 |
| Entropy | 82.23 | 76.73 | 81.13 |
| BANSA (D) | 82.43 | 76.91 | 81.33 |
| BANSA (B) | **82.56** | **78.06** | **81.66** |

Table 8: **Effect of varying the number of $K$.**

| $K$ | Subject Consistency | Background Consistnecy |
|---|---|---|
| 1 | 0.9618 | 0.9788 |
| 3 | 0.9623 | 0.9793 |
| 5 | 0.9632 | 0.9798 |
| 7 | 0.9638 | 0.9802 |
| 10 | **0.9641** | **0.9811** |

Table 9: **Quantitative comparison of reversed BANSA scoring on CogVideoX-2B.** This presents results when selecting samples using the highest BANSA scores, compared to the default selection.

| Method | Subject Consistency | Background Consistency | Temporal Flickering | Motion Smoothness | Aesthetic Quality | Imaging Quality | Dynamic Degree | Quality Score |
|---|---|---|---|---|---|---|---|---|
| Vanilla | 0.9616 | 0.9788 | 0.9715 | 0.9743 | 0.6195 | 0.6267 | 0.6380 | 82.08 |
| + Ours (reverse) | 0.9626 | 0.9785 | 0.9700 | 0.9741 | 0.6181 | 0.6253 | 0.6328 | 81.93 |
| + Ours | **0.9641** | **0.9811** | **0.9775** | **0.9746** | **0.6202** | **0.6276** | **0.6511** | **82.56** |

Table 10: **Component analysis of Bernoulli masking parameters on Wan2.1 1.3B.**

| Backbone Model | Bernoulli $p$ | Subject Consistency ↑ | Motion Smoothness ↑ | Aesthetic Quality ↑ | Imaging Quality ↑ | Overall Consistency ↑ |
|---|---|---|---|---|---|---|
| | 0.0 | 0.9254 | 0.9751 | 0.6464 | 0.6459 | 0.2731 |
| | 0.2 | **0.9310** | **0.9786** | **0.6559** | 0.6516 | **0.2763** |
| Wan2.1 1.3B | 0.5 | 0.9308 | 0.9779 | 0.6543 | **0.6533** | 0.2727 |
| | 0.7 | 0.9302 | 0.9778 | 0.6557 | 0.6529 | 0.2726 |
| | 1.0 | 0.9232 | 0.9753 | 0.6451 | 0.6508 | 0.2721 |

Table 11: **Quantitative evaluation on the IPOC-2B after post-training and Wan2.1 14B model.**

| Backbone Model | Method | Subject Consistency ↑ | Background Consistency ↑ | Temporal Flickering ↑ | Motion Smoothness ↑ | Aesthetic Quality ↑ | Imaging Quality ↑ | Dynamic Degree ↑ |
|---|---|---|---|---|---|---|---|---|
| IPOC-2B | Vanilla | 0.9273 | 0.9535 | 0.9918 | 0.9741 | 0.6554 | 0.664 | 0.7527 |
| | + Ours | **0.9281** | **0.9534** | **0.9921** | **0.9745** | **0.6561** | **0.6644** | **0.7721** |
| Wan2.1 14B | Vanilla | 0.9290 | 0.9587 | 0.9931 | 0.9759 | 0.6572 | **0.6552** | 0.5916 |
| | + Ours | **0.9302** | **0.9596** | **0.9938** | **0.9772** | **0.6601** | 0.6529 | **0.6277** |

**Effect of Noise Pool Size $M$.** We analyze the effect of noise pool size $M$, which controls the diversity of candidate seeds in BANSA. Larger $M$ improves the chance of finding high-quality seeds but increases inference cost. As shown in Figure 8 (Appendix), performance saturates around $M = 10$ for CogVideoX-2B and -5B, which we adopt as the default for each model.

**Reversing the BANSA Criterion.** To further validate BANSA, we conduct a control experiment where the noise seed with the *highest* BANSA score is selected—i.e., choosing the seed associated with the greatest model uncertainty. As shown in Table 9, this reversal results in degradation of quality-related metrics, confirming that lower BANSA scores are predictive of perceptually stronger generations and supporting the validity of our selection strategy.

**Effect of Bernoulli masking probability $p$.** We conduct a sensitivity analysis on the Bernoulli masking probability $p$ using the Wanx 2.1 (1.3B) model. As shown in Table 10, both disabling masking ($p = 0$) and fully random masking ($p = 1$) degrade performance, indicating that the masking probability has a meaningful impact on the results. The method remains stable within the intermediate range $p \in [0.2, 0.7]$, where overall performance varies only slightly. Among these configurations, $p = 0.2$ delivers the best performance, which supports our choice of the default setting.

**Evaluation on strong video backbones and post-RL models.** We further evaluate our method on stronger settings, including the large-capacity Wan2.1 14B backbone and the post-trained IPOC-2B model. IPOC-2B is refined with a DPO-related preference alignment method (Yang et al., 2025), making it a competitive post-RL baseline. As shown in Table 11, our method consistently improves over the vanilla models across most VBench dimensions, demonstrating generalization to larger architectures and robustness under preference-aligned post-RL training.

**User Study.** We conducted a human preference study in which evaluators compared paired videos generated with and without our method along two criteria: overall quality and text–video alignment. As shown in Fig. 5 (a), our method is consistently preferred on both aspects, indicating improvements in perceptual quality and semantic alignment. The details are provided in Appendix K.

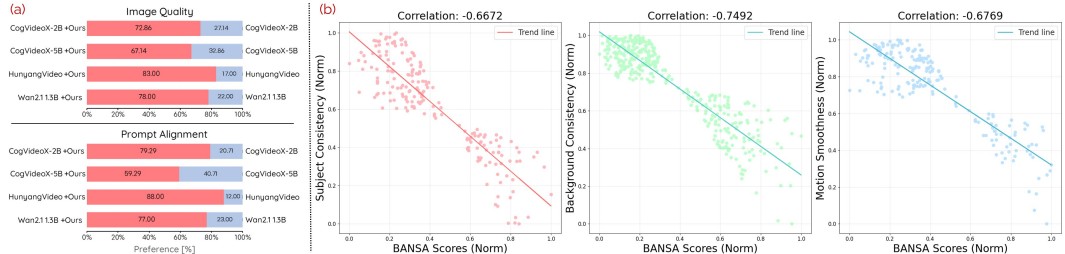

Figure 5: (a) User study showing consistent human preference for our method in overall quality and text–video alignment. (b) Correlation analysis where BANSA scores exhibit strong negative trends with key VBench metrics, indicating that lower scores correspond to higher-quality generations.

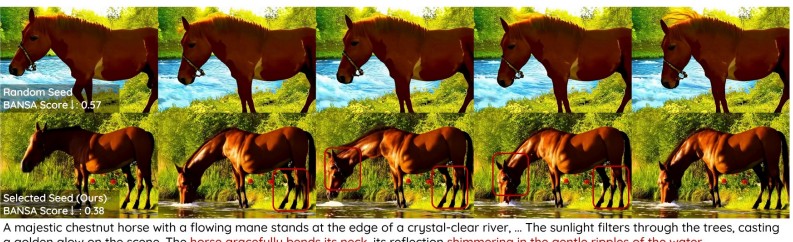

A majestic chestnut horse with a flowing mane stands at the edge of a crystal-clear river, ... The sunlight filters through the trees, casting a golden glow on the scene. The horse gracefully bends its neck, its reflection shimmering in the gentle ripples of the water.

Figure 6: **Failure case and limitation of our method.** Although the BANSA score indicates low uncertainty, the resulting video still contains unnatural content. This represent a limitation of ours: we select optimal seeds but do not alter the generation process itself.

**Analysis on correlation with actual quality.** We examined how BANSA scores relate to quality by sampling diverse prompts and noise inputs and visualizing their score–quality trends in Fig. 5(b). Pearson correlations show strong negative relationships with Subject Consistency (–0.6672), Background Consistency (–0.7492), and Motion Smoothness (–0.6769), indicating that lower BANSA scores reliably associate with higher-quality and more stable generations.

# 7    DISCUSSION AND LIMITATIONS

Our method focuses on noise–seed selection based on model uncertainty, which introduces several limitations. As shown in Figure 6, even seeds with low BANSA scores may still produce unnatural results because ANSE does not modify the subsequent sampling process. Moreover, BANSA captures uncertainty at the attention level, which does not fully account for semantic or aesthetic quality. Although evaluating multiple candidates with stronger quality metrics could reduce this gap, such an approach is computationally prohibitive. A promising direction for future work is to integrate ANSE with post-training refinement methods—such as Self-Forcing (SF) (Huang et al., 2025)—that directly enhance sample quality. Notably, ANSE already works effectively with post-RL refinement methods like IPoC, and extending this compatibility to SF represents a natural next step. Such combinations may yield complementary gains in both quality and robustness beyond seed selection alone.

# 8    CONCLUSION

We present ANSE, a framework for active noise selection in video diffusion models. At its core is BANSA, an acquisition function that leverages attention-derived uncertainty to identify noise seeds yielding confident, consistent attention and thus higher-quality generations. BANSA adapts the BALD principle to the generative setting by operating in attention space, with efficient deployment enabled through Bernoulli-masked attention and lightweight layer selection. Experiments across multiple T2V backbones show that ANSE improves video quality and prompt alignment with minimal inference overhead. This introduces an inference-time scaling paradigm, enhancing generation not by altering the model or sampling steps, but through informed seed selection.

ACKNOWLEDGEMENT

This work was partially supported by Institute of Information & Communications Technology Planning & Evaluation (IITP) grant funded by the Korean government (MSIT) (Artificial Intelligence Graduate School Program (GIST)) (No. 2019-0-01842).

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

## A SUPPLEMENTARY SECTION

In this supplementary document, we present the following:

- Proof of Proposition 1 from the main paper regarding the BANSA Zero condition in Section B.

- Implementation details of ANSE and evaluation metrics in Section C.

- Further explanation of layer selection through cumulative BANSA correlation in Section D.

- Additional ablation studies, including full-layer BANSA score analysis (Section E), temporal scope effects (Section F), and attention-space effects (Section G).

- Additional analysis, including effect of CFG scale (Section H), equivalent computation budget (Section I), statistical validation (Section J), and user study protocol (Section K).

- Additional qualitative results demonstrating the impact of BANSA Score in Section L.

## B PROOF OF PROPOSITION 1

**Proposition 1** (BANSA Zero Condition). *Let $\mathcal{A}(\mathbf{z}, c, t) = \{A^{(1)}, \ldots, A^{(K)}\}$ be a set of row-stochastic attention maps. Then:*

$$BANSA(\mathbf{z}, c, t) = 0 \quad \Leftrightarrow \quad A^{(1)} = \cdots = A^{(K)}.$$

*Proof.* BANSA is defined as the difference between the average entropy and the entropy of the average:

$$\text{BANSA}(\mathbf{z}, c, t) = \mathcal{H}\left(\frac{1}{K}\sum_{k=1}^{K} A^{(k)}(\mathbf{z}, c, t)\right) - \frac{1}{K}\sum_{k=1}^{K}\mathcal{H}(A^{(k)}(\mathbf{z}, c, t)).$$

Since the Shannon entropy $\mathcal{H}(\cdot)$ is strictly concave over the probability simplex. Therefore, by Jensen's inequality:

$$\mathcal{H}\left(\frac{1}{K}\sum_{k=1}^{K} A^{(k)}\right) \geq \frac{1}{K}\sum_{k=1}^{K}\mathcal{H}(A^{(k)}),$$

with equality if and only if $A^{(1)} = \cdots = A^{(K)}$. Thus, BANSA$(\mathbf{z}, c, t) = 0$ if and only if all attention maps are identical. □

**Remark 1.** *(Interpretation) This result confirms that the BANSA score quantifies disagreement among sampled attention maps. A BANSA score of zero occurs only when all stochastic attention realizations collapse to a single deterministic map—i.e., the model exhibits **no epistemic uncertainty** in its attention distribution. Higher BANSA values indicate greater variation across samples, and thus, higher uncertainty. In this sense, BANSA acts as a Jensen–Shannon-type divergence over attention maps, capturing their dispersion under stochastic masking.*

## C FURTHER DETAILS ON EVALUATION METRICS AND IMPLEMENTATION

**Evaluation Metrics** To evaluate performance on Vbench, we use the Vbench-long version, where prompts are augmented using GPT-4o across all evaluation dimensions. This version is specifically designed for assessing videos longer than 4 seconds.

We rigorously evaluate our generated videos following the official evaluation protocol. The Quality Score is a weighted average of the following aspects: subject consistency, background consistency, temporal flickering, motion smoothness, aesthetic quality, imaging quality, and dynamic degree.

The Semantic Score is a weighted average of the following semantic dimensions: object class, multiple objects, human action, color, spatial relationship, scene, appearance style, temporal style, and overall consistency.

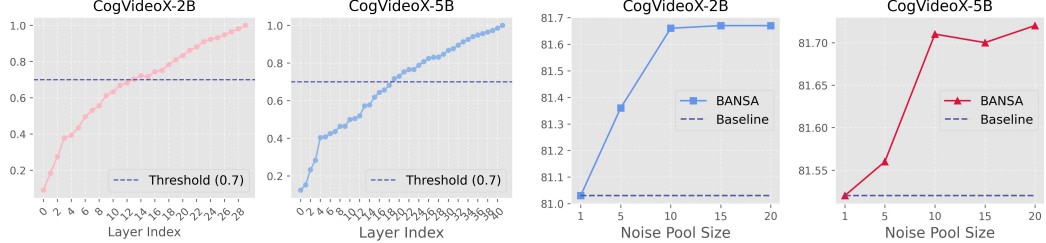

Figure 7: **Correlation analysis between cumulative BANSA score and full-layer scores.** The 0.7 threshold is reached around layer 14 for CogVideoX-2B and layer 19 for CogVideoX-5B.

Figure 8: **Ablation study on noise pool size $M$.** We evaluate total scores across three text-to-video diffusion models with varying $M$, and select suitable values based on computational cost.

The Total Score is then computed as a weighted combination of the Quality Score and Semantic Score:

$$\text{Total Score} = \frac{w_1}{w_1 + w_2} \cdot \text{Quality Score} + \frac{w_2}{w_1 + w_2} \cdot \text{Semantic Score}$$

where $w_1 = 4$ and $w_2 = 1$, following the default setting in the official implementation.

**Implementation**   We compute our BANSA score using the BALD-style formulation, which yields non-negative values. For clearer visualization, we normalize the BANSA scores from their original range (minimum: 0.45, maximum: 0.60) to the [0, 1] interval. This normalization is used solely for visual clarity in figures and plots, and does not affect the noise selection process, which operates on the raw BANSA scores. All models are evaluated using their default inference configurations: AnimateDiff (50 steps), CogVideo-X (50 steps), Wan2.1 (50 steps), and HunyuanVideo (50 steps).

## D    FURTHER DETAIL OF BANSA LAYER-WISE CORRELATION ANALYSIS

**Prompt construction.**   We evenly sampled 100 prompts from the four official VBench categories: *Subject Consistency*, *Overall Consistency*, *Temporal Flickering*, and *Scene*. Each category contains 25 prompts, selected to ensure diversity in motion, structure, and semantics.

Below are representative examples:

- *Subject Consistency* (e.g., "A young man with long, flowing hair sits on a rustic wooden stool in a cozy room, strumming an acoustic guitar...")
- *Overall Consistency* (e.g., "A mesmerizing splash of turquoise water erupts in extreme slow motion, each droplet suspended in mid-air...")
- *Temporal Flickering* (e.g., "A cozy restaurant with flickering candles and soft music. Patrons dine peacefully as snow falls outside...")
- *Scene* (e.g., "A university campus transitions from lively student life to a golden sunset behind the clock tower...")

Prompt sampling was stratified to ensure coverage of diverse visual and temporal patterns. The full list of prompts will be made publicly available upon code release.

**BANSA score computation and correlation analysis.**   For each prompt, we generated 10 videos using different random noise seeds and computed BANSA scores at each attention layer. This yielded one full-layer BANSA score and a set of layer-wise scores per seed. To obtain stable estimates, we averaged the per-layer and full-layer BANSA scores across the 10 seeds, reducing noise-specific variance and capturing consistent uncertainty patterns.

We then computed Pearson correlations between the cumulative BANSA scores (summed from layer 1 to $d$) and the official quality scores. The optimal depth $d^*$ was defined as the smallest $d$ at which the correlation exceeded 0.7, a widely used threshold indicating a strong positive relationship. As shown in Figure 7, this point was reached at $d^* = 14$ for CogVideoX-2B and $d^* = 19$ for CogVideoX-5B.

For AnimateDiff, the correlation crossed the threshold earlier at $d^* = 10$, while for HunyuanVideo and Wanx 2.1 it consistently appeared around $d^* = 20$. These model-specific depths were applied throughout all experiments. Such consistency across architectures empirically supports our choice of the 0.7 threshold and ensures that the correlation analysis reflects generalizable, noise-agnostic patterns of attention-based uncertainty.

Table 12: Comparison between full-layer and truncated BANSA score.

| Backbone Model | Method | Subject Consistency ↑ | Background Consistency ↑ | Temporal Flickering ↑ | Motion Smoothness ↑ | Aesthetic Quality ↑ | Imaging Quality ↑ | Dynamic Degree ↑ | Quality Score ↑ | Inference Time ↓ |
|---|---|---|---|---|---|---|---|---|---|---|
| CogvideoX-2B | Full-layer | 0.9639 | 0.9810 | **0.9801** | 0.9743 | 0.6198 | 0.6244 | **0.6516** | **82.58** | 303.7 |
| | Truncated | **0.9641** | **0.9811** | 0.9775 | **0.9746** | **0.6202** | **0.6276** | 0.6511 | 82.56 | **269.3** |
| CogvideoX-5B | Full-layer | **0.9660** | 0.9630 | **0.9863** | 0.9708 | 0.6168 | 0.6290 | **0.6979** | **82.71** | 810.3 |
| | Truncated | 0.9658 | 0.9639 | 0.9861 | **0.9711** | **0.6179** | 0.6290 | 0.6918 | 82.70 | **754.3** |

# E  EFFECTIVENESS OF TRUNCATED BANSA SCORE

To reduce the computational overhead of BANSA evaluation, we adopt a truncated score that aggregates attention uncertainty only up to a fixed depth $d^*$, rather than summing over all layers. To evaluate the effectiveness of this approximation, we compared the final generation quality when selecting noise seeds using either the full-layer or truncated BANSA scores.

As shown in Table 12, both approaches yield highly similar results across all seven dimensions of the VBench evaluation protocol (subject consistency, background consistency, aesthetic quality, imaging quality, motion smoothness, dynamic degree, and temporal flickering). Importantly, the overall quality scores are preserved despite the substantial reduction in attention layers used.

This demonstrates that truncated BANSA is sufficient to capture the key uncertainty signals for reliable noise selection while reducing inference time. The strong alignment in quality stems from the fact that our method relies on relative ranking rather than absolute values, allowing for efficient yet robust selection with significantly lower computational cost. We attribute this effectiveness to the fact that most informative attention behaviors emerge early in the denoising process, allowing accurate uncertainty estimation without full-layer computation.

Table 13: Effect of temporal scope in BANSA score on generation quality.

| BANSA Scope | Subject Consistency ↑ | Temporal Flickering ↑ | Motion Smoothness ↑ | Aesthetic Quality ↑ | Imaging Quality ↑ | Dynamic Degree ↑ | Inference Time ↓ |
|---|---|---|---|---|---|---|---|
| 1-step | 0.9639 | **0.9801** | 0.9743 | 0.6198 | 0.6244 | **0.6516** | × 1 |
| 25-step avg | 0.9651 | 0.9798 | 0.9746 | 0.6202 | 0.6271 | 0.6511 | × 25 |
| 50-step avg | **0.9652** | 0.9799 | **0.9751** | **0.6203** | **0.6276** | 0.6514 | × 50 |

# F  EFFECT OF TEMPORAL SCOPE IN BANSA SCORE

While our method computes the BANSA score only at the first denoising step to minimize cost, it is natural to ask whether incorporating more timesteps improves its predictive power for noise selection. To investigate this, we compute the average BANSA score across the first 1, 25, and 50 denoising steps and compare their effectiveness in predicting video quality.

Table 13 reports the VBench scores for subject consistency, aesthetic quality, imaging quality, motion smoothness, dynamic degree, and temporal flickering when using BANSA computed over different temporal scopes. Although using more timesteps results in slightly better quality, the gains are marginal. This indicates that most of the predictive signal for noise quality is embedded early in the generation trajectory.

More importantly, since BANSA is used solely to assess the uncertainty of the initial noise seed—not to track full-step generation behavior—our 1-step computation is sufficient to capture the core uncertainty signal. In contrast, computing BANSA over all steps requires running multiple attention forward passes across the full trajectory, resulting in substantial computational overhead that limits its practicality for real-world applications.

Table 14: **Quantitative results on VBench using AnimateDiff, CogVideoX-2B and -5B.**

| Backbone Model | Method | Quality Score | Semantic Score | Total Score |
|---|---|---|---|---|
| AnimateDiff (Guo et al., 2024b) | Vanilla | 80.22 | 69.03 | 77.98 |
| | + Ours (Self-Attention) | **81.68** | **70.72** | **79.19** |
| | + Ours (Cross-Attention) | **81.66** | **71.09** | **79.33** |

## G   EFFECT OF ATTENTION SPACE IN BANSA SCORE

To further examine whether BANSA scores capture quality- or semantic-related uncertainty, we conducted additional experiments with the AnimateDiff backbone, which separates self- and cross-attention layers. Unlike CogVideo and Wanx, which adopt an MMDiT architecture where multimodal attention entangles both components, AnimateDiff provides a clearer lens for disentangling semantic effects. As shown in Table 14, applying BANSA to self-attention primarily improved perceptual quality scores, whereas applying it to cross-attention enhanced semantic alignment. This indicates that uncertainty over noise manifests differently depending on the attention type, supporting the interpretation that BANSA reflects both quality- and semantics-related variations. In contrast, MMDiT-based models such as CogVideo and Wanx integrate these dimensions within their unified attention structure, making them more suitable for overall noise selection in the main paper's experiments.

Table 15: **Ablation study on the interaction between higher CFG scale and ours on Wan2.1(1.3B).**

| Backbone Model | Method | CFG Scale | Subject Consistency ↑ | Motion Smoothness ↑ | Aesthetic Quality ↑ | Imaging Quality ↑ |
|---|---|---|---|---|---|---|
| Wanx 2.1 1.3B | Vanilla | 5.0 | 0.9285 | 0.9748 | 0.6465 | **0.6519** |
| | Ours | 5.0 | **0.9310** | **0.9786** | **0.6559** | 0.6516 |
| | Vanilla | 7.5 | 0.9266 | 0.9740 | 0.6447 | 0.6357 |
| | Ours | 7.5 | **0.9276** | **0.9755** | **0.6519** | **0.6409** |

## H   EFFECT OF CFG SCALE ON OUR METHOD

Classifier-free guidance (CFG) modifies the conditional–unconditional interpolation only at the final stage of inference, whereas our method selects noise seeds at initialization based on attention-level uncertainty. Since they operate at different points in the pipeline, their effects are largely independent: CFG adjusts the extrapolation strength, while our method chooses seeds that lead to more stable early-stage behavior.

To assess potential interactions, we increased the CFG scale from the default value of 5.0 to 7.5 on the Wanx 2.1 (1.3B) model, as shown in Table 15. Higher CFG values generally degrade performance by pushing sampling beyond the model's calibrated range. Nevertheless, our method consistently outperforms the vanilla baseline under both settings, indicating that its benefits remain stable across different CFG scales.

Table 16: Ablation study under an equivalent computation budget. Increasing the baseline sampling steps does not consistently improve performance, whereas our method achieves better results with fewer effective steps.

| Backbone Model | Method | Steps | Subject Consistency ↑ | Motion Smoothness ↑ | Aesthetic Quality ↑ | Imaging Quality ↑ |
|---|---|---|---|---|---|---|
| Wan2.1 1.3B | Vanilla | 50 | 0.9285 | 0.9748 | 0.6465 | **0.6519** |
| | Vanilla | 60 | 0.9274 | 0.9745 | 0.6476 | 0.6492 |
| | Ours | < 60 | **0.9310** | **0.9786** | **0.6559** | 0.6516 |

## I    EFFECT OF EQUIVALENT COMPUTATION BUDGET

Our method evaluates $M = 10$ noise candidates, but the actual computation is lower than $50 + 10$ steps because many candidates terminate early during partial denoising. To assess whether the baseline could benefit from additional computation, we increased its sampling steps from 50 to 60. As shown in Table 16, this results in only marginal changes and even degrades some metrics, likely because the model is not calibrated for longer sampling schedules.

In contrast, our method consistently improves performance while requiring fewer effective steps than the 60-step baseline, suggesting that the gains stem from selecting more stable noise initializations rather than from additional compute. This highlights the efficiency advantage of our approach under equivalent or lower computation budgets.

Table 17: Comparison of VBench Quality Scores With and Without BANSA. Scores are reported as mean (95% confidence interval, CI: lower – upper).

| Backbone Model | Method | Subject Consistency ↑ (CI) | Background Consistency ↑ (CI) | Temporal Flickering ↑ (CI) | Motion Smoothness ↑ (CI) | Aesthetic Quality ↑ (CI) | Imaging Quality ↑ (CI) | Dynamic Degree ↑ (CI) |
|---|---|---|---|---|---|---|---|---|
| CogvideoX-2B | Baseline | 0.9616 (0.9606 - 0.9626) | 0.9788 (0.9779 - 0.9797) | 0.9715 (0.9688 - 0.9742) | 0.9738 (0.9733 - 0.9753) | 0.6195 (0.6116 - 0.6274) | **0.6267 (0.6160 - 0.6374)** | 0.6380 (0.6133 - 0.6627) |
| | + Ours | **0.9639 (0.9629 - 0.9649)** | **0.9810 (0.9795 - 0.9825)** | **0.9801 (0.9781 - 0.9821)** | **0.9743 (0.9737 - 0.9756)** | **0.6198 (0.6119 - 0.6277)** | 0.6244 (0.6137 - 0.6351) | **0.6500 (0.6255 - 0.6745)** |
| CogvideoX-5B | Baseline | 0.9616 (0.9608 - 0.9624) | 0.9616 (0.9602 - 0.9630) | 0.9862 (0.9833 - 0.9891) | **0.9712 (0.9703 - 0.9721)** | 0.6162 (0.6084 - 0.6240) | 0.6272 (0.6170 - 0.6374) | 0.6895 (0.6655 - 0.7135) |
| | + Ours | **0.9660 (0.9653 - 0.9667)** | **0.9630 (0.9616 - 0.9644)** | **0.9863 (0.9835 - 0.9891)** | 0.9708 (0.9700 - 0.9716) | **0.6168 (0.6089 - 0.6247)** | **0.6290 (0.6189 - 0.6391)** | **0.6979 (0.6738 - 0.7220)** |

Table 18: Comparison of VBench Semantic Scores With and Without BANSA. Scores are reported as mean (95% confidence interval, CI: lower – upper).

| Backbone Model | Method | Object Class ↑ (CI) | Multiple Object ↑ (CI) | Color ↑(CI) | Spatial Relationship ↑( CI) | Scene ↑ (CI) | Temporal Style ↑ (CI) | Overall Consistency ↑ (CI) | Human Action ↑ (CI) | Appearance Style↑ (CI) |
|---|---|---|---|---|---|---|---|---|---|---|
| CogvideoX-2B | Baseline | 0.8522 (0.8511 - 0.8533) | 0.6391 (0.6376 - 0.6406) | **0.8318 (0.8302 - 0.8334)** | **0.7148 (0.6963 - 0.7333)** | 0.4815 (0.4801 - 0.4829) | 0.2488 (0.2453 - 0.2523) | 0.2722 (0.2679 - 0.2765) | 0.9860 (0.9743 - 0.9977) | 0.2497 (0.2483 - 0.2511) |
| | + Ours | **0.8842 (0.8831 - 0.8853)** | **0.6596 (0.6581 - 0.6611)** | 0.8255 (0.8240 - 0.8270) | 0.6915 (0.6728 - 0.7102) | **0.5286 (0.5272 - 0.5300)** | **0.2547 (0.2511 - 0.2583)** | **0.2760 (0.2718 - 0.2802)** | **0.9870 (0.9757 - 0.9963)** | **0.2507 (0.2493 - 0.2521)** |
| CogvideoX-5B | Baseline | 0.8595 (0.8585 - 0.8605) | 0.6511 (0.6496 - 0.6526) | **0.8218 (0.8202 - 0.8234)** | **0.6872 (0.6675 - 0.7069)** | 0.5233 (0.5219 - 0.5247) | 0.2552 (0.2513 - 0.2591) | 0.2761 (0.2717 - 0.2805) | 0.9900 (0.9813 - 0.9987) | 0.2483 (0.2354 - 0.2612) |
| | + Ours | **0.8675 (0.8665 - 0.8685)** | **0.6658 (0.6643 - 0.6673)** | 0.8201 (0.8186 - 0.8216) | 0.6831 (0.6637 - 0.7025) | **0.5255 (0.5241 - 0.5269)** | **0.2585 (0.2546 - 0.2624)** | **0.2773 (0.2730 - 0.2816)** | **0.9980 (0.9910 - 1.0050)** | **0.2524 (0.2511 - 0.2537)** |

Table 19: Comparison of VBench Quality Scores for HunyuanVideo and Wan2.1. Scores are reported as mean (95% confidence interval, CI: lower – upper).

| Backbone Model | Method | Subject Consistency ↑ (CI) | Background Consistency ↑ (CI) | Motion Smoothness ↑ (CI) | Aesthetic Quality ↑ (CI) | Imaging Quality ↑ (CI) | Temporal Flickering ↑ (CI) |
|---|---|---|---|---|---|---|---|
| Hunyuan Video | Vanilla | 0.9562 (0.9551 − 0.9573) | 0.9656 (0.9645 − 0.9667) | 0.9850 (0.9838 − 0.9862) | **0.6276 (0.6198 − 0.6354)** | 0.6137 (0.6059 − 0.6215) | 0.9921 (0.9902 − 0.9940) |
| | + Ours | **0.9612 (0.9603 − 0.9621)** | **0.9661 (0.9650 − 0.9672)** | **0.9858 (0.9847 − 0.9869)** | 0.6268 (0.6191 − 0.6345) | **0.6151 (0.6075 − 0.6227)** | **0.9938 (0.9919 − 0.9957)** |
| Wan2.1 | Vanilla | 0.9562 (0.9553 − 0.9571) | 0.9656 (0.9644 − 0.9668) | 0.9850 (0.9839 − 0.9861) | **0.6276 (0.6201 − 0.6351)** | 0.6137 (0.6061 − 0.6213) | 0.9943 (0.9924 − 0.9962) |
| | + Ours | **0.9612 (0.9601 − 0.9623)** | **0.9661 (0.9649 − 0.9673)** | **0.9858 (0.9846 − 0.9870)** | 0.6268 (0.6190 − 0.6346) | **0.6151 (0.6073 − 0.6229)** | **0.9956 (0.9935 − 0.9977)** |

## J    STATISTICAL VALIDATION OF RESULTS

Our VBench evaluation uses 4,750 videos (five samples per prompt) across 17 fine-grained dimensions, where even small absolute gains are meaningful. To assess statistical significance, we computed 95% confidence intervals using 10 independent runs, each based on random subsets of 100 videos. As shown in Tables 17, 18, and 19, our method consistently improves or maintains performance across quality-, semantic-, and motion-related metrics without exhibiting bias toward any particular subset. These statistical results provide strong evidence of the robustness and reliability of our approach.

## K    USER STUDY PROTOCOL

As presented in Fig. 5 (a), we conduct a human evaluation study to complement automated metrics and directly assess whether ANSE improves video quality and prompt coherence. We evaluate four representative video diffusion models—CogVideoX-2B, CogVideoX-5B, Wan 2.1, and Hunyuan-Video—by comparing their outputs with and without ANSE. To construct a fair evaluation protocol, we randomly sample 30 prompts from the *Subject Consistency* and *Overall Consistency* dimensions of the VBench benchmark. For each prompt, we generate paired videos using identical noise seeds for both the baseline and the ANSE-equipped models.

We recruit 12 independent human evaluators, all of whom remain fully blind to the models and are restricted to participating only once. Each evaluator is shown two videos side-by-side (baseline vs. ANSE) and responds to two questions:

1. **Video Quality:** "Which video appears more stable and visually pleasing?"

2. **Prompt Alignment:** "Which video better represents the given text prompt?"

The order of prompts and the arrangement of video pairs are fully randomized. Across all backbones and prompts, models equipped with ANSE are consistently preferred in terms of both video quality and prompt alignment, demonstrating the practical effectiveness and robustness of ANSE.

## L  ADDITIONAL QUALITATIVE COMPARISON

**More qualitative results.**  Figures 9, 10, 11 12, and 13 present additional examples generated using our noise selection framework including diverse backbone such as, AnimateDiff, CogVideoX2B and 5B, HunyangVideo, and Wan2.1. Across diverse prompts, the selected seeds yield improved spatial detail, aesthetic quality, and semantic alignment, further validating the robustness of our approach. These examples complement our quantitative findings by illustrating the visual impact of BANSA-based noise selection.

**Effect of BANSA score on generation quality.**  Figures 14 provide a qualitative comparison of outputs generated using three types of noise seeds: a randomly sampled seed, the seed with the highest BANSA score (lowest quality), and the seed with the lowest BANSA score (highest quality). All videos were generated using 50 denoising steps with the CogVideoX-5B backbone. The lowest-BANSA seed consistently produces sharper, more coherent, and semantically faithful videos, whereas the highest-BANSA seed often leads to structural artifacts or temporal instability. These results highlight the practical value of BANSA-guided noise selection.

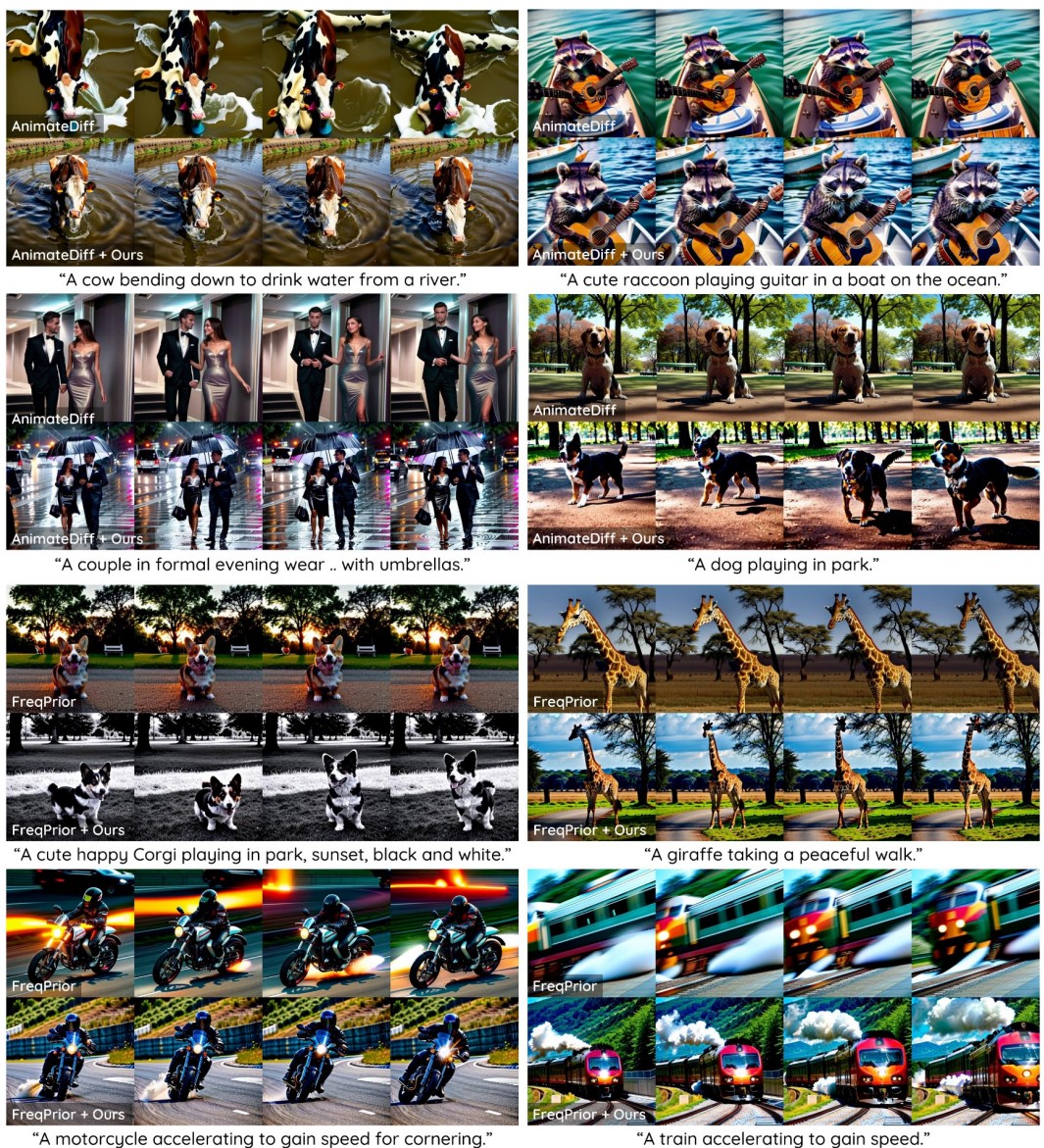

Figure 9: **Effect of ANSE on semantic fidelity and motion stability in AnimateDiff and FreqPrior.** Each block compares baseline generations with those using ANSE-selected noise. In addition, while FreqPrior serves as a noise-refinement baseline, our method is fully compatible and achieves further improvements when combined.

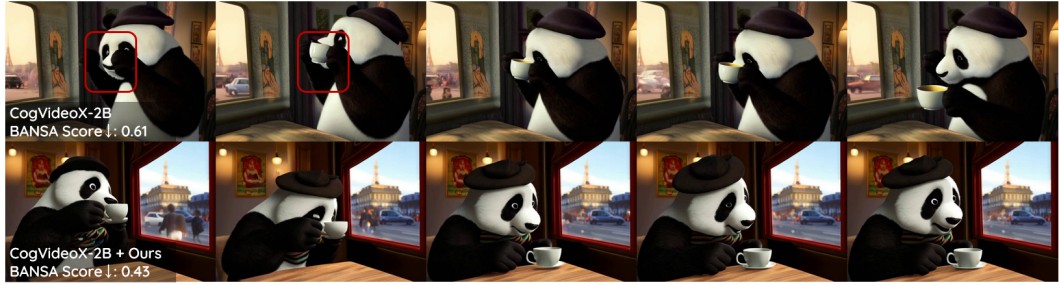

"In a charming Parisian café, an animated panda sits at a quaint wooden table, sipping coffee from a delicate porcelain cup. The panda, wearing a stylish beret and a striped scarf, gazes out the window at the bustling Paris streets,...."

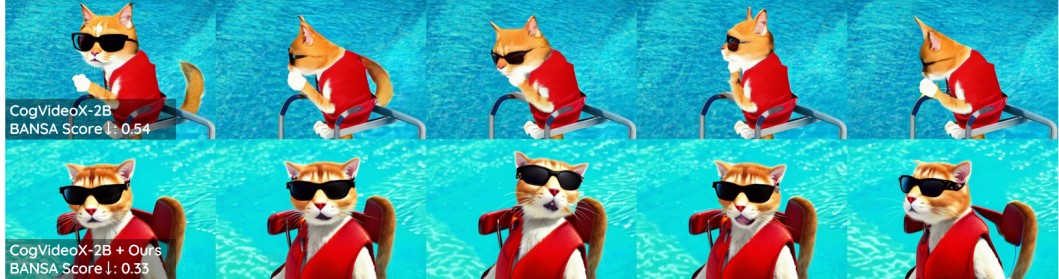

"A cool cat, sporting sleek black sunglasses and a red lifeguard vest, sits confidently on a high lifeguard chair overlooking a sparkling blue pool. The feline's fur is a mix of orange and white, and its tail flicks with authority.."

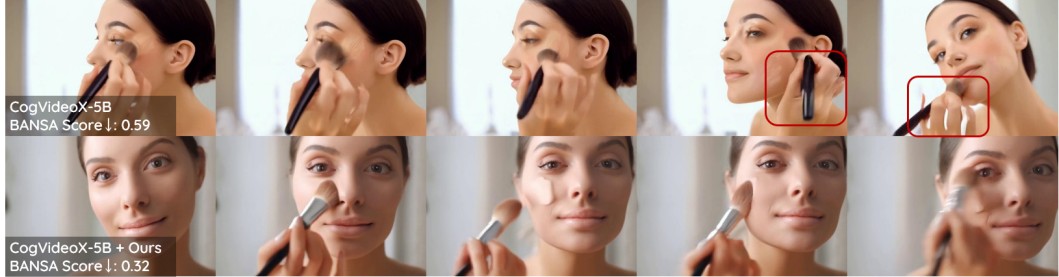

"A young woman with flawless skin and a serene expression sits at a vanity, bathed in soft morning light. She begins by applying a light moisturizer, her fingers moving gently across her face. Next, she uses a foundation brush to blend a sheer.."

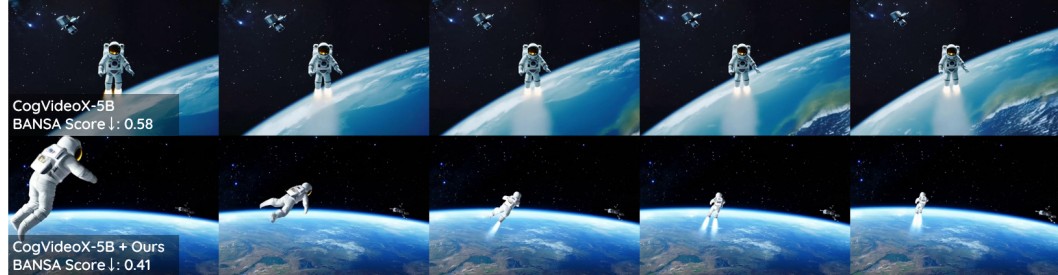

"An astronaut in a pristine white spacesuit, floats effortlessly against the vast, star-studded expanse of space. As the camera zooms out, the intricate details of the suit, including the life-support backpack and tether,."

Figure 10: **Effect of ANSE on semantic fidelity and motion stability in CogVideoX outputs.** Each block compares baseline generations with those using ANSE-selected noise. Across both CogVideoX-2B and 5B, ANSE improves semantic alignment to the prompt and reduces artifacts such as temporal flickering and object distortion. White numbers show BANSA scores for the same prompt.

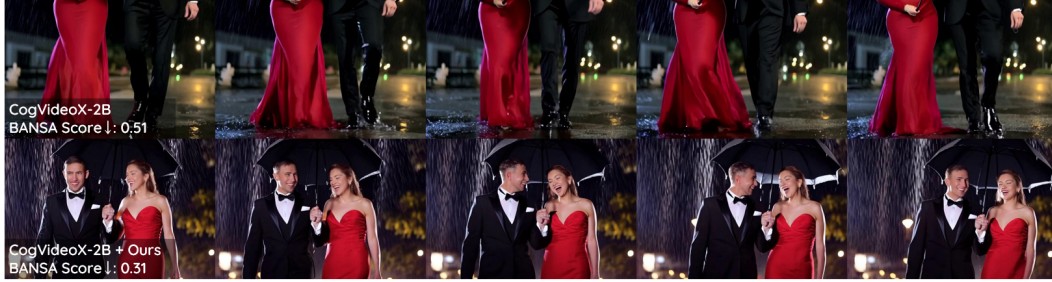

"A sophisticated couple, dressed in elegant evening attire, walks down a dimly lit street, their formal. The man, in a tailored black tuxedo, and the woman, in a flowing red gown, share a black umbrella as the rain captures their synchronized steps.."

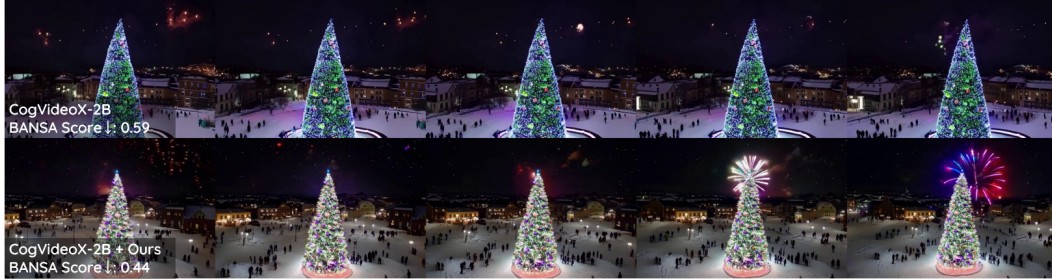

"A drone captures a breathtaking aerial view of a festive celebration in a snow-covered town square, centered around a towering, brilliantly lit Christmas tree adorned with twinkling lights and ornaments. The scene is alive with vibrant fireworks.."

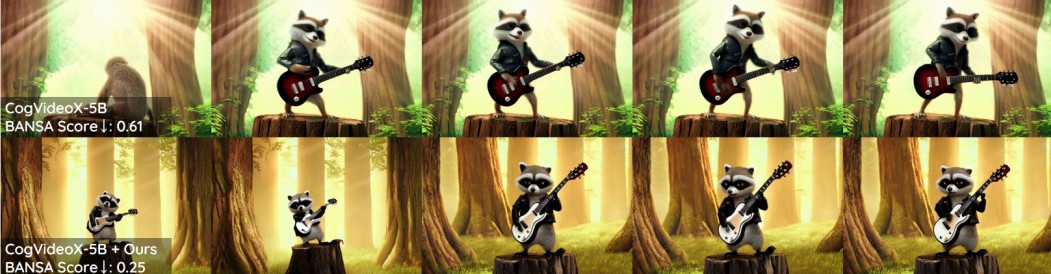

"In a whimsical forest clearing, a raccoon with a mischievous glint in its eye stands on a tree stump, holding an electric guitar. The raccoon, wearing a tiny leather jacket, strums the guitar with surprising skill, its tiny paws moving deftly over the strings...."

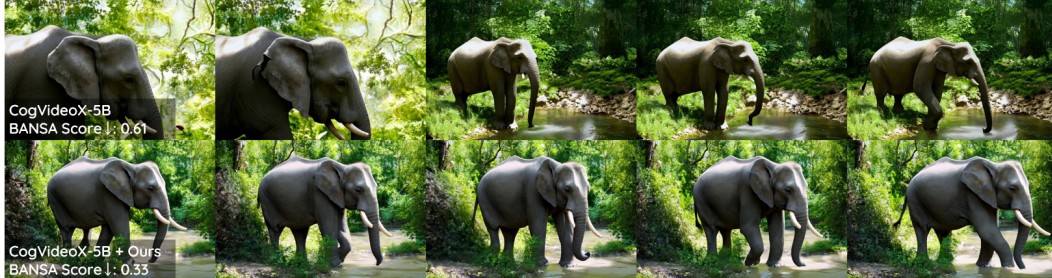

"A majestic elephant strolls gracefully through a lush, verdant forest, its massive feet gently pressing into the soft earth. The sunlight filters through the dense canopy, casting dappled shadows on its wrinkled, grey skin.... ...."

Figure 11: **Additional qualitative comparison of CogVideoX variants with and without ANSE.** Results from CogVideoX-2B are shown in the first two rows; the rest show CogVideoX-5B. With ANSE, videos exhibit improved visual quality, better text alignment, and smoother motion transitions compared to the baseline. White numbers show BANSA scores for the same prompt.

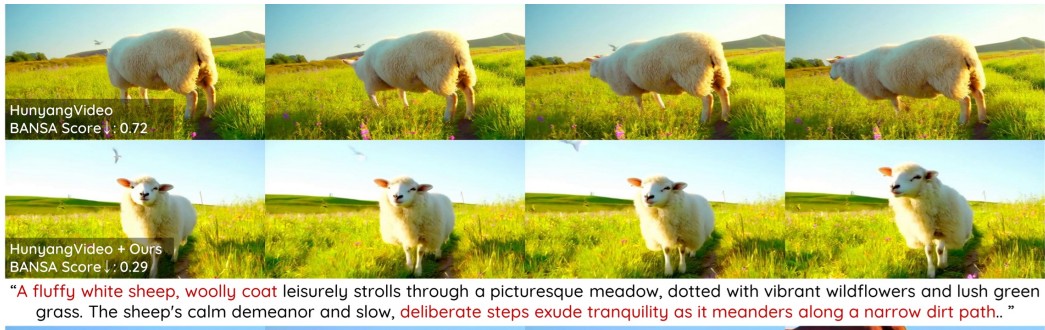

"A fluffy white sheep, woolly coat leisurely strolls through a picturesque meadow, dotted with vibrant wildflowers and lush green grass. The sheep's calm demeanor and slow, deliberate steps exude tranquility as it meanders along a narrow dirt path.. "

" A majestic chestnut horse with a glossy coat leisurely strolls through a sun-dappled meadow, its mane gently swaying in the breeze. As it walks, the horse's hooves softly tread on the lush, green grass, creating a rhythmic,.."

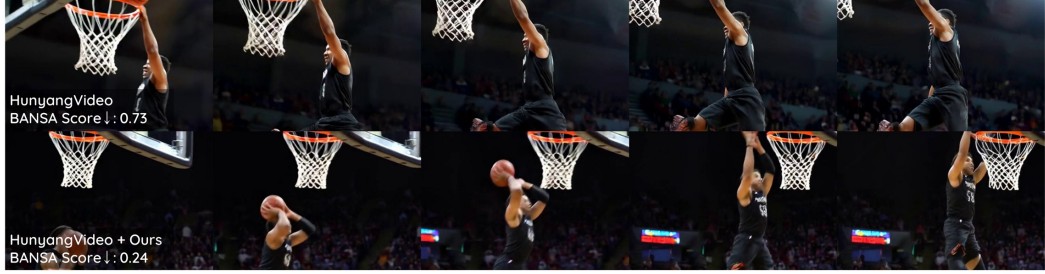

"A dynamic athlete, soars through the air in a packed, electrifying arena. with sweat glistening on their determined face, grips the basketball tightly. As they approach the hoop, emphasizing grace of the jump.

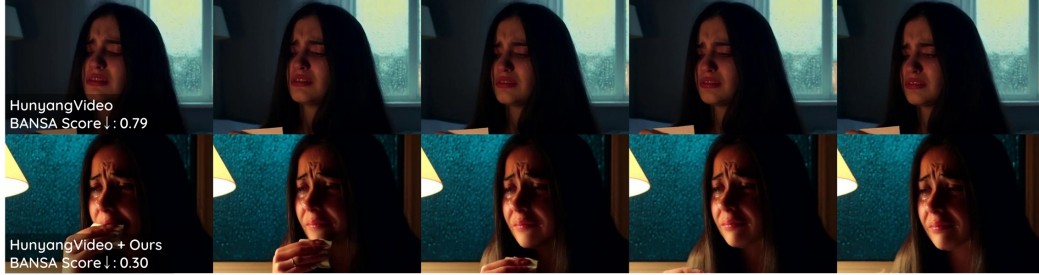

"A young woman with long, dark hair sits alone in a dimly lit room, her face illuminated by the soft glow of a nearby lamp. Tears stream down her cheeks, ….The camera captures her quivering lips and the silent sobs that shake her shoulders.., "

Figure 12: **Additional qualitative comparison of HunyangVideo with and without ANSE.** With ANSE, the generated videos achieve sharper visual fidelity, stronger alignment between text and content, and smoother motion progression than the baseline. White numbers show BANSA scores for the same prompt.

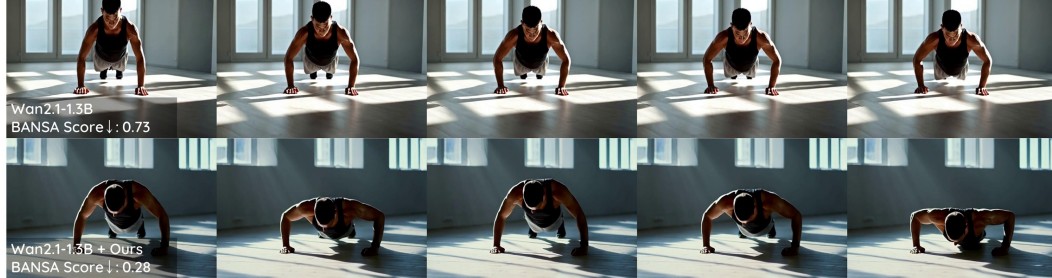

"A determined individual in a sleek black tank top and gray athletic shorts performs push-ups on a pristine wooden floor in a minimalist, sunlit room. The camera captures the sweat glistening on their forehead, emphasizing their intense focus .."

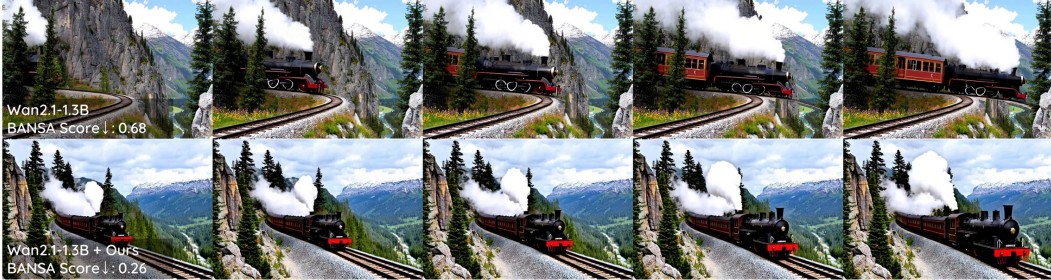

"A majestic steam train, with its vintage black and red carriages, chugs along a winding mountainside track, enveloped in a cloud of white steam. The train's powerful engine, adorned with brass accents, gleams in the sunlight as it ascends...."

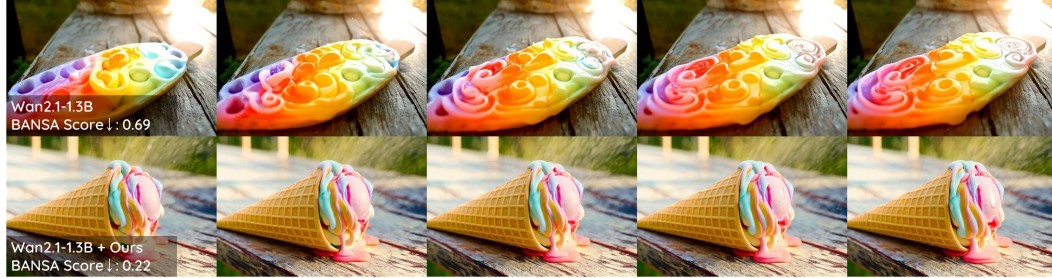

"A vibrant, multi-colored ice cream cone sits on a rustic wooden table, its creamy swirls beginning to soften under the warm sunlight streaming through a nearby window. The camera zooms in to capture the intricate details of the melting ice cream"

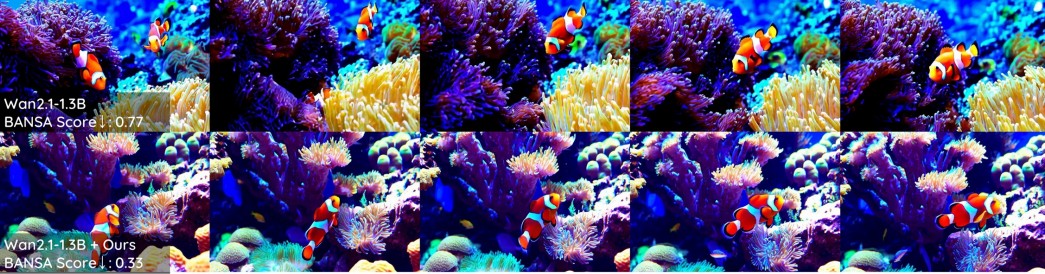

"A vibrant clownfish, with its striking orange and white stripes, gracefully navigates through a lush coral reef teeming with life. The fish weaves between the intricate branches of colorful corals, which range from deep purples to bright yellows, ...."

Figure 13: **Additional qualitative comparison of Wan2.1 with and without ANSE.** ANSE enables videos to deliver higher visual quality, more accurate adherence to the given text, and more seamless motion dynamics compared to the baseline. White numbers show BANSA scores for the same prompt.

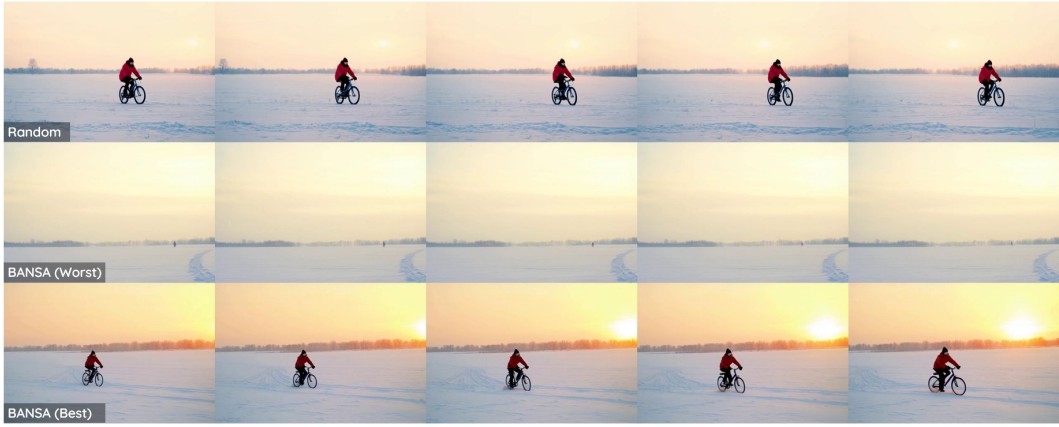

"A lone bicycle, with its sleek frame and black tires, glides effortlessly through a vast, snow-covered field under a pale winter sky. The rider, bundled in a red parka, black gloves, and a woolen hat, pedals steadily, leaving a delicate trail in the pristine snow. The scene captures the quiet serenity of the landscape, with snowflakes gently falling and the distant silhouette of bare trees lining the horizon. As the rider continues, the sun

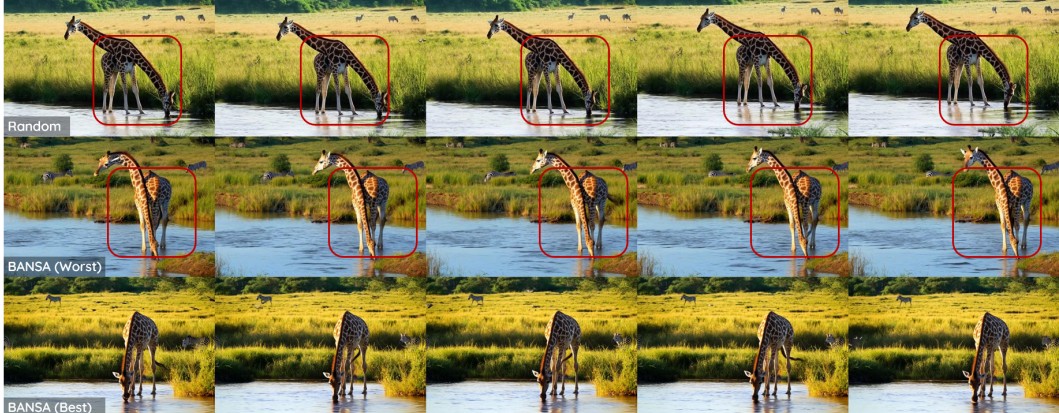

"A majestic giraffe, its long neck gracefully arching, bends down to drink from a serene river, surrounded by lush greenery and tall grasses. The sun casts a golden glow, highlighting the giraffe's patterned coat and the gentle ripples in the water. Nearby, a family of zebras grazes peacefully, adding to the tranquil scene. Birds flutter above, their reflections dancing on the water's surface. The giraffe's delicate movements create a sense of harmony with nature, as the river flows gently, reflecting the vibrant colors of the surrounding landscape..."

Figure 14: **Qualitative comparison of generations from different noise seeds.** We compare outputs generated from a randomly sampled seed (top), the seed with the highest BANSA score (middle), and the seed with the lowest score (bottom), using the same prompt and model. BANSA-selected seeds produce more coherent structure, stable motion, and stronger semantic alignment than both random and high-uncertainty seeds.

