# OpenReview forum: "Model Already Knows the Best Noise: Bayesian Active Noise Selection via Attention  in Video Diffusion Model"
_ICLR.cc/2026/Conference — ICLR 2026 Poster_

### Official Review · Reviewer_YgYi · 2025-10-28

**Soundness:** 3
**Presentation:** 4
**Contribution:** 3
**Rating:** 6
**Confidence:** 4

**Summary:**

The paper presents ANSE, a framework for active noise selection in video diffusion models using BANSA, an attention-based acquisition function that identifies noise seeds yielding consistent and confident generations. BANSA adapts the BALD principle to attention space with lightweight, efficient inference. Experiments across multiple text-to-video backbones show that ANSE may enhance video quality and prompt alignment with slight overhead.

**Strengths:**

* The paper is clearly written.

* The accompanying website includes excellent figures and animations, which are much appreciated.

* The experiments and ablation studies are detailed and thorough.

**Weaknesses:**

1. **Ensemble Size**
   - In Table 7, the metrics consistently improve as the ensemble size $K$ increases.
   - Could the authors provide results with even larger $K$ values?

2. **Denoising Steps**
   - What is the number of denoising steps used for each model?
   - It appears to be 50 based on the appendix, but this is only explicitly mentioned for the CogVideoX-5B backbone (L839).

3. **Computation Budget (NFE)**
   - From a neural function evaluation (NFE) perspective, the total cost is effectively $M + 50$ (or the actual number of denoising steps).
   - What would happen if the baseline methods were also given an equivalent $M + 50$ computation budget?

4. **Ablation on Bernoulli-Masked Attention**
   - Please include ablation results for the Bernoulli-masked attention mechanism, particularly analyzing the effect of the masking probability parameter $p = 0.2$.

5. **Evaluation Metrics**
   - Consider including additional motion-specific metrics, such as FVMD [1], to better assess the claimed improvements.
   - The sample videos on the website look good and visibly outperform the baseline. A human evaluation could further validate these qualitative improvements, as VBench—while comprehensive—may have limitations in capturing perceptual quality.

Ref:\
[1] Liu, J., Qu, Y., Yan, Q., Zeng, X., Wang, L. and Liao, R., 2024. Fr'echet Video Motion Distance: A Metric for Evaluating Motion Consistency in Videos. arXiv preprint arXiv:2407.16124.

**Questions:**

Please see the [Weakness] section.

---

> ### Author Response · Authors · 2025-11-19
> **Author Response (1/2).**
>
> We thank Reviewer YgYi for the constructive feedback and for recognizing the clarity of our writing, the thoroughness of our ablation studies, and the quality of our figures and animations.
> Below are point-by-point summaries of the reviewer’s main comments.
>
> **[YgYi] requested evaluation with larger $K$ values.**
>
> > We thank the reviewer for this helpful suggestion. In response, we extended the evaluation beyond our default setting of $K=10$ and tested larger values ($K=20$ and $K=50$), as shown in **Table R1**. While increasing $K$ slightly improves background consistency, the gains are marginal and do not increase monotonically across all metrics. Considering the additional computation cost that scales linearly with $K$, we found that $K=10$ provides the best trade-off between performance and efficiency. We have updated the manuscript accordingly and appreciate the reviewer’s thoughtful comment.
>
> **Table R1. Effect of varying the number of $K$**
>
> | **Backbone Model** | **K** | **Subject Consistency ↑** | **Background Consistency ↑** |
> |---|---|---|---|
> | **CogVideoX-2B** | 1 | 0.9681 | 0.9788 |
> |  | 3 | 0.9623 | 0.9793 |
> |  | 5 | 0.9632 | 0.9798 |
> |  | 7 | 0.9638 | 0.9802 |
> |  | 10 | 0.9641 | 0.9811 |
> |  | 20 | 0.9638 | 0.9809 |
> |  | 50 | 0.9645 | 0.9813 |
>
> **[YgYi] asked about the number of denoising steps used for each model.**
>
> > We thank the reviewer for the question. All models are evaluated using their default inference configurations: AnimateDiff (50 steps), CogVideo-X (50 steps), Wan2.1 (50 steps), and HunyuanVideo (50 steps). We have clarified these settings in **the revised manuscript (Appendix C)**, and we appreciate the reviewer’s suggestion.
>
>
> **[YgYi] inquired what would happen if the baseline methods were given an equivalent computation budget.**
>
> > Thanks for the helpful question. Although we use $M=10$ noise candidates, the actual computation cost is below $50+10$ steps because many candidates terminate early during partial denoising. To examine whether the baseline benefits from additional computation, we increased its sampling steps from 50 to 60. As shown in **Table R2 (Table 16 in the revised manuscript)**, this adjustment produces only minor changes and even degrades some metrics, likely because the model is not tuned for longer schedules. In contrast, our method still improves performance while requiring fewer effective steps than the 60-step baseline, indicating that the gain comes from selecting a more stable noise initialization rather than increased computation. We appreciate the reviewer’s concrete comment and have updated the manuscript accordingly.
>
>
> **Table R2. Ablation study under an equivalent computation budget**
> Increasing the baseline sampling steps does not consistently improve performance, whereas our method achieves better results with fewer effective steps.
>
> | **Backbone Model** | **Method** | **Steps** | **Subject Consistency ↑** | **Motion Smoothness ↑** | **Aesthetic Quality ↑** | **Imaging Quality ↑** |
> |---|---|---|---|---|---|---|
> | **Wan2.1 1.3B** | Vanilla | 50 | 0.9285 | 0.9748 | 0.6465 | **0.6519** |
> |  | Vanilla | 60 | 0.9274 | 0.9745 | 0.6476 | 0.6492 |
> |  | Ours | < 60 | **0.9310** | **0.9786** | **0.6559** | 0.6516 |

---

> ### Author Response · Authors · 2025-11-19
> **Author Response (2/2).**
>
> **[YgYi] requested analysis of the masking probability parameter.**
>
> > Per the reviewer’s suggestion, we conducted a sensitivity study on the Bernoulli masking probability $p$ using the Wanx 2.1 (1.3B) model, as shown in **Table R3 (Table 10 in the revised manuscript)**. The results show that disabling masking entirely ($p=0$) or applying fully random masking ($p=1$) leads to noticeable degradation, indicating that the masking probability does influence overall quality. At the same time, the intermediate range $p \in [0.2, 0.7]$ produces similar results, suggesting that the method is largely robust away from boundary cases. Among these settings, $p=0.2$ provides the best trade-off, which motivates our choice of the default configuration. We sincerely appreciate this insightful suggestion and have incorporated the corresponding analyses into the revised manuscript.
>
> **Table R3. Component analysis of Bernoulli masking parameters on Wan2.1 1.3B**
>
> | **Backbone Model** | **Bernoulli p** | **Subject Consistency ↑** | **Motion Smoothness ↑** | **Aesthetic Quality ↑** | **Imaging Quality ↑** | **Overall Consistency ↑** |
> |---|---|---|---|---|---|---|
> | **Wan2.1 1.3B** | 0.0 | 0.9254 | 0.9751 | 0.6464 | 0.6459 | 0.2731 |
> |  | 0.2 | **0.9310** | **0.9786** | **0.6559** | 0.6516 | **0.2763** |
> |  | 0.5 | 0.9308 | 0.9779 | 0.6543 | **0.6533** | 0.2727 |
> |  | 0.7 | 0.9302 | 0.9778 | 0.6557 | 0.6529 | 0.2726 |
> |  | 1.0 | 0.9232 | 0.9753 | 0.6451 | 0.6508 | 0.2721 |
>
>
> **[YgYi] suggested including additional motion-specific metrics, such as FVMD.**
>
> > We thank the reviewer for the valuable suggestion to include motion-specific metrics. Accordingly, we conducted additional experiments comparing the Fréchet Video Motion Distance (FVMD) on the MSR-VTT dataset [1] using HunyuanVideo and Wan2.1 (1.3B) as base models, as shown in **Table R4 (Table 6 of revised manuscripts)**. We randomly selected 200 text prompts from the MSR-VTT test set and generated videos using both the baseline models and our method. The corresponding 200 reference videos were used to compute FVMD. We observe that our method yields improved FVMD scores compared to the baselines. This further validates our improvements in motion quality, consistent with our VBench findings. These results have been included in the revised manuscript.
>
> [1] Xu, Jun et al. “MSR-VTT: A large video description dataset for bridging video and language.” CVPR 2016.
>
> **Table R4. Comparison with FVMD metrics on MSRVTT dataset**
>
> | **Backbone Model** | **Methods** | **FVMD ↓** |
> |---|---|---|
> | **HunyuanVideo** | Vanilla | 17641.19 |
> |  | + Ours | **16491.68** |
> | **Wan2.1 (1.3B)** | Vanilla | 16495.59 |
> |  | + Ours | **14306.19** |
>
> **[YgYi] requested results from a human evaluation study.**
>
> > Thanks for the constructive suggestion. Following your request, we conducted a human preference study comparing results with and without our method on CogVideoX-2B, CogVideoX-5B, HunyuanVideo, and Wan2.1, and the results are now included in **the revised manuscript (Figure 5(a))**. Human evaluators assessed paired videos generated from the same prompts based on two criteria: overall visual quality and text–video alignment. For each model, we prepared a representative set of evaluation pairs and collected responses from 12 independent raters, which was sufficient to reveal clear and consistent preference trends. Across all models and both criteria, our method was consistently preferred, demonstrating improved perceptual quality and semantic faithfulness.

---

> > ### Comment · Reviewer_YgYi · 2025-11-24
> >
> > Dear Authors,
> >
> > Thank you very much for your dedicated efforts to address my concerns. Most of them have been resolved, and I will raise my rating.
> >
> > One more slight request: could you add more details about the user study protocols in the appendix (writing) for completeness?
> >
> > It would also be interesting to see whether the proposed method could be applied to combined autoregressive+diffusion style video generation models such as Self-Forcing, as a preliminary study to broaden the impact of the work.

---

> > > ### Author Response · Authors · 2025-11-24
> > >
> > > Dear Reviewer [YgYi],
> > >
> > > Thank you very much for your thoughtful follow-up feedback and for kindly raising your rating. We truly appreciate your time and consideration.
> > >
> > > Per your request, we have updated the appendix with additional details on the user study protocol for completeness.
> > >
> > > Regarding your suggestion on combined autoregressive + diffusion models such as Self-Forcing, we sincerely appreciate this insightful direction. Self-Forcing appears to be a post-trained extension of Wan2.1, and following Reviewer [KJHH]'s  earlier request, we have already demonstrated in Table 11 that ANSE is applicable to post-trained video diffusion models in the RL setting. Given this, applying ANSE to Self-Forcing is technically feasible, and we plan to explore this integration in future experiments. We have also added this point as a promising future research direction in the Discussion section.
> > >
> > > Thank you again for the constructive suggestions and for your positive reassessment.

---

### Official Review · Reviewer_KjHH · 2025-10-30

**Soundness:** 2
**Presentation:** 4
**Contribution:** 3
**Rating:** 4
**Confidence:** 5

**Summary:**

This paper addresses noise selection in video diffusion models by introducing a metric called BANSA, which utilizes attention information to estimate model confidence and consistency. It also proposes a Bernoulli-masked approximation to efficiently compute scores from a single diffusion step.

**Strengths:**

+ A novel metric is defined to guide noise selection.
+ The paper is well-organized and presents the idea clearly.

**Weaknesses:**

- Would the method remain effective when applied to a powerful base video model (e.g., WAN 2.2/2.1 14B), or when the video model has undergone post-training procedures such as RL or DPO?
- There is a lack of visualizations showing the score distributions for different prompts and samples from different noise inputs.
- How does the Bernoulli masking affect the score distribution? Is there a significant difference between using and not using the mask?
- It would be helpful to show generated samples from the same prompt but with different scores, to help validate the effectiveness of the proposed metric.
- What is the underlying reason that using this score leads to improvements in motion, composition, and artifact reduction? More insight into the mechanism would be valuable.

**Questions:**

Please refer to the strengths and weaknesses

---

> ### Author Response · Authors · 2025-11-19
> **Author Response (1/2).**
>
> We thank Reviewer KjHH for the constructive feedback and valuable suggestions.
> Below are point-by-point summaries of the reviewer’s main questions.
>
>
> **[KjHH] requested evaluation on a stronger base video model (e.g., Wan 2.1 14B) or one trained with post-RL procedures.**
>
> > Following the reviewer’s suggestion, we conducted additional evaluations on two stronger backbones: Wan2.1 14B and IPOC-2B after post-training, as shown in **Table R1 and R2 (Table 11 of the revised manuscript)**. The IPOC-2B model is refined via Iterative Preference Optimization (IPO) [1], a preference-based training method related to Direct Preference Optimization (DPO) and point-wise scoring approaches such as Kahneman–Tversky Optimization. This makes IPOC-2B a substantially stronger and more reward-aligned backbone. Across both models, integrating our method consistently improves the vanilla baseline on most VBench dimensions, indicating that our approach generalizes well and remains robust even when applied to larger-capacity or preference-optimized models. We sincerely appreciate this valuable suggestion, which enabled us to perform a stronger validation and have updated the manuscript accordingly.
>
> [1] Yang, Xiaomeng, Zhiyu Tan, and Hao Li. “IPO: Iterative preference optimization for text-to-video generation.” arXiv:2502.02088
>
> **Table R1. Quantitative evaluation on the Wan2.1 14B model**
> Our method consistently improves the baseline across most VBench dimensions.
>
> | **Backbone Model** | **Method** | **Subject Consistency ↑** | **Background Consistency ↑** | **Temporal Flickering ↑** | **Motion Smoothness ↑** | **Aesthetic Quality ↑** | **Imaging Quality ↑** | **Dynamic Degree ↑** |
> |---|---|---|---|---|---|---|---|---|
> | **Wan2.1 14B** | Vanilla | 0.9290 | 0.9587 | 0.9931 | 0.9759 | 0.6572 | **0.6552** | 0.5916 |
> |  | + Ours | **0.9302** | **0.9596** | **0.9938** | **0.9772** | **0.6601** | 0.6529 | **0.6277** |
>
>
> **Table R2. Quantitative evaluation on IPOC-2B after post-training via Iterative Preference Optimization (IPO)**
> Our method provides consistent improvements across VBench metrics.
>
> | **Backbone Model** | **Method** | **Subject Consistency ↑** | **Background Consistency ↑** | **Temporal Flickering ↑** | **Motion Smoothness ↑** | **Aesthetic Quality ↑** | **Imaging Quality ↑** | **Dynamic Degree ↑** |
> |---|---|---|---|---|---|---|---|---|
> | **IPOC-2B** | Vanilla | 0.9273 | 0.9535 | 0.9918 | 0.9741 | 0.6554 | 0.6640 | 0.7527 |
> |  | + Ours | **0.9281** | **0.9534** | **0.9921** | **0.9745** | **0.6561** | **0.6644** | **0.7721** |
>
> **[KjHH] noted the absence of visualizations showing score distributions across different prompts and noise inputs.**
>
> > We thank the reviewer for this valuable suggestion. To better illustrate how BANSA scores relate to actual generation quality, we randomly sampled diverse prompts and noise inputs and plotted BANSA scores against the corresponding VBench metrics. These visualizations have been added to the main paper for improved clarity. To further quantify this relationship, we computed the Pearson correlation between BANSA scores and key VBench dimensions. We observed strong negative correlations with Subject Consistency (–0.6672), Background Consistency (–0.7492), and Motion Smoothness (–0.6769), indicating that lower BANSA scores reliably correspond to higher-quality generations, as shown in **Figure 5(b) of the revised manuscript**. This supports our central hypothesis that BANSA effectively identifies noise seeds that lead to better semantic and temporal coherence. We sincerely appreciate the reviewer’s insightful comment, which helped us strengthen both the empirical validation and the overall presentation of the paper.

---

> ### Author Response · Authors · 2025-11-19
> **Author Response (2/2).**
>
> **[KjHH] asked how Bernoulli masking affects quality and whether it causes significant changes.**
>
> > Per the reviewer’s suggestion, we conducted a sensitivity study on the Bernoulli masking probability $p$ using the Wanx 2.1 (1.3B) model, as shown in **Table R3 (Table 10 in the revised manuscript)**. The results show that disabling masking entirely ($p=0$) or applying fully random masking ($p=1$) leads to noticeable degradation, indicating that the masking probability does influence overall quality. At the same time, the intermediate range $p \in [0.2, 0.7]$ produces similar results, suggesting that the method is largely robust away from boundary cases. Among these settings, $p=0.2$ provides the best trade-off, which motivates our choice of the default configuration. We sincerely appreciate this insightful suggestion and have incorporated the corresponding analyses into the revised manuscript.
>
> **Table R3. Component analysis of Bernoulli masking parameters on Wan2.1 1.3B**
>
> | **Backbone Model** | **Bernoulli p** | **Subject Consistency ↑** | **Motion Smoothness ↑** | **Aesthetic Quality ↑** | **Imaging Quality ↑** | **Overall Consistency ↑** |
> |---|---|---|---|---|---|---|
> | **Wan2.1 1.3B** | 0.0 | 0.9254 | 0.9751 | 0.6464 | 0.6459 | 0.2731 |
> |  | 0.2 | **0.9310** | **0.9786** | **0.6559** | 0.6516 | **0.2763** |
> |  | 0.5 | 0.9308 | 0.9779 | 0.6543 | **0.6533** | 0.2727 |
> |  | 0.7 | 0.9302 | 0.9778 | 0.6557 | 0.6529 | 0.2726 |
> |  | 1.0 | 0.9232 | 0.9753 | 0.6451 | 0.6508 | 0.2721 |
>
>
>
> **[KjHH] suggested showing generated samples from the same prompt but with different scores to validate the proposed metric.**
>
> > We thank the reviewer for this helpful suggestion. The reviewer is kindly reminded that all qualitative figures were generated using the same text prompts and models, with the corresponding BANSA scores explicitly annotated. These examples already demonstrate how samples with higher and lower BANSA scores differ in perceptual alignment. To further improve clarity, we will highlight this correspondence more explicitly in the revised version, making the relationship between the BANSA score and generation quality clearer.
>
>
> **[KjHH] inquired about why using the proposed score improves motion, composition, and artifact reduction, and requested more insight into the mechanism.**
>
> > We thank the reviewer for this insightful question. As shown in **Table 2 of the main paper**, samples with higher BANSA scores exhibit more stable and consistent attention patterns, while **Table 3** demonstrates that they also show smaller latent-trajectory variation across timesteps. These findings suggest that BANSA tends to select noise seeds that induce smoother temporal evolution and more coherent spatial layouts, thereby reducing artifacts and improving motion consistency. This observation is consistent with prior studies such as FreeInit and FreqPrior, which similarly report that stable latent initialization leads to higher-quality generations. Taken together, these results provide empirical evidence that BANSA improves generation quality by promoting stable and semantically consistent latent dynamics.

---

> > ### Comment · Reviewer_KjHH · 2025-11-24
> >
> > Thanks for the replies. The improvement seems quite marginal in the three tables. Is any other way to validate the proposed method quantitatively?

---

> > > ### Author Response · Authors · 2025-11-24
> > >
> > > We thank Reviewer `KjHH` for raising this point. As suggested by Reviewer `YgYi`, we conducted additional experiments using the MSR-VTT dataset [1] and evaluated motion quality with the Fréchet Video Motion Distance (FVMD) metric [2]. We randomly sampled 200 prompts from the MSR-VTT test split and generated paired videos with HunyuanVideo and Wan 2.1 (1.3B). Each generated video was paired with its corresponding reference video to compute FVMD.
> > >
> > > The results, shown below in **Table R4 (also Table 6 in the revised manuscript)**, indicate that our method consistently achieves lower FVMD scores across both backbones, demonstrating improved motion fidelity.
> > >
> > > **Table R4. Comparison with FVMD metrics on the MSR-VTT dataset**
> > >
> > > | **Backbone Model** | **Methods** | **FVMD ↓** |
> > > |--------------------|------------|-----------|
> > > | **HunyuanVideo**   | Vanilla    | 17641.19  |
> > > |                    | + Ours     | **16491.68** |
> > > | **Wan2.1 (1.3B)**  | Vanilla    | 16495.59  |
> > > |                    | + Ours     | **14306.19** |
> > >
> > > While we acknowledge that some VBench dimension gains may appear marginal, FVMD is reported to correlate more strongly with human perception than standard automated motion metrics. Under this more human-aligned measure, the observed reductions represent meaningful improvements in motion consistency and align well with our VBench findings.
> > >
> > > To further validate the perceptual impact of our method, we also conducted a **human preference study** with 12 independent blind evaluators. As shown in **Figure 5(a)** and detailed in the updated appendix, evaluators consistently preferred videos generated with our method in terms of both visual quality and prompt alignment.
> > >
> > > We thank the reviewer once again for the constructive suggestion and for giving us the opportunity to strengthen our quantitative and perceptual validation.
> > >
> > > [1] Xu *et al.* “MSR-VTT: A Large Video Description Dataset for Bridging Video and Language.” CVPR 2016.
> > > [2] Liu *et al.* “Fréchet Video Motion Distance: A Metric for Evaluating Motion Consistency in Videos.” ICML 2024 Workshop on Uncertainty in Generative AI.

---

> > > > ### Comment · Reviewer_KjHH · 2025-11-24
> > > >
> > > > Thanks for the additional statement. My concerns are addressed. I will increase my rating.

---

> > > > > ### Author Response · Authors · 2025-11-24
> > > > >
> > > > > We truly appreciate your reconsideration, and we’re glad that our clarification addressed your concerns.
> > > > > Thanks again for taking the time to improve the rating.

---

### Official Review · Reviewer_CXmX · 2025-10-30

**Soundness:** 2
**Presentation:** 2
**Contribution:** 2
**Rating:** 6
**Confidence:** 2

**Summary:**

The paper proposes ANSE (Active Noise Selection for Generation), an inference-time framework that improves text-to-video diffusion by selecting the initial noise seed that minimizes a new attention-based epistemic uncertainty measure, BANSA (Bayesian Active Noise Selection via Attention). This method improves video quality and temporal coherence across various T2V backbones and is approximated efficiently via Bernoulli-masked attention and layer truncation chosen by correlation analysis to keep overhead low.

**Strengths:**

1. The central idea of using attention-based uncertainty as an internal signal to guide noise selection is innovative. It shifts the paradigm from relying on external, often heuristic-based priors to a principled method that is aware of the model's own confidence.
2. Through approximations (Bernoulli masking, layer truncation), authors make the consistent quality gains with a reasonable inference overhead. Demonstrates robust generalization and real-world viability.

**Weaknesses:**

1. Lack of statistical confidence reporting. Improvements are modest in absolute terms; without standard deviations/CI or significance tests, it’s hard to assess robustness and effect sizes across prompts/seeds.
2. Lack of Sensitivity Analysis for Key Hyperparameters. The method relies on several key hyperparameters, notably the Bernoulli masking probability p (set to 0.2) . The paper does not provide a sensitivity analysis for these values. It is unclear how the optimal choices were determined and how robust the method's performance is to variations in these parameters.

**Questions:**

1. The paper mentions that BANSA can be applied to cross-, self-, or temporal attention. Did the authors investigate which type of attention map is most predictive of final video quality?
2. The Classifier-Free Guidance (CFG) scale is a crucial parameter for controlling prompt alignment in diffusion models. How does the choice of noise seed, as guided by BANSA, interact with different CFG values? Is it possible that a high BANSA score could be improved by using a higher CFG scale, or does the initial noise fundamentally constrain the quality regardless of the guidance strength?

---

> ### Author Response · Authors · 2025-11-19
> **Author Response (1/2).**
>
> We thank the reviewer for recognizing our method as innovative and for acknowledging its paradigm-shifting potential through a principled approach and robust generalization.
>
> **[CXmX] required statistical significance analysis.**
> > We thank the reviewer for this valuable suggestion. Our main VBench evaluation uses 4,750 videos (five samples per prompt) across 17 fine-grained dimensions, where even small absolute gains correspond to meaningful improvements.
> To further assess statistical significance, we computed 95% confidence intervals using 10 independent runs, each based on random subsets of 100 videos.
> As shown in **Tables R1–R3 (17–19 in the main paper)**, our method consistently improves or maintains performance across quality-, semantic-, and motion-related metrics (e.g., Motion Smoothness, Dynamic Degree), without bias toward specific subsets.
> We have updated the relevant analyses in the revised supplemental material. We appreciate the reviewer’s comment, which helped us strengthen empirical rigor; the corresponding results will be reflected in the final manuscript (Appendix J).
>
> **Table R1. Comparison of VBench Quality Scores With and Without BANSA**
> Scores are reported as mean (95% confidence interval, CI: lower – upper).
>
> | **Backbone Model** | **Method** | **Subject Consistency ↑ (CI)** | **Background Consistency ↑ (CI)** | **Temporal Flickering ↑ (CI)** | **Motion Smoothness ↑ (CI)** | **Aesthetic Quality ↑ (CI)** | **Imaging Quality ↑ (CI)** | **Dynamic Degree ↑ (CI)** |
> |---|---|---|---|---|---|---|---|---|
> | **CogVideoX-2B** | Baseline | 0.9616 (0.9606–0.9626) | 0.9788 (0.9779–0.9797) | 0.9715 (0.9688–0.9742) | 0.9738 (0.9733–0.9753) | 0.6195 (0.6116–0.6274) | **0.6267 (0.6160–0.6374)** | 0.6380 (0.6133–0.6627) |
> |  | + Ours | **0.9639 (0.9629–0.9649)** | **0.9810 (0.9795–0.9825)** | **0.9801 (0.9781–0.9821)** | **0.9743 (0.9737–0.9756)** | **0.6198 (0.6119–0.6277)** | 0.6244 (0.6137–0.6351) | **0.6500 (0.6255–0.6745)** |
> | **CogVideoX-5B** | Baseline | 0.9616 (0.9608–0.9624) | 0.9616 (0.9602–0.9630) | 0.9862 (0.9833–0.9891) | **0.9712 (0.9703–0.9721)** | 0.6162 (0.6084–0.6240) | 0.6272 (0.6170–0.6374) | 0.6895 (0.6655–0.7135) |
> |  | + Ours | **0.9660 (0.9653–0.9667)** | **0.9630 (0.9616–0.9644)** | **0.9863 (0.9835–0.9891)** | 0.9708 (0.9700–0.9716) | **0.6168 (0.6089–0.6247)** | **0.6290 (0.6189–0.6391)** | **0.6979 (0.6738–0.7220)** |
>
> ---
>
> **Table R2. Comparison of VBench Semantic Scores With and Without BANSA**
>
> | **Backbone Model** | **Method** | **Object Class ↑ (CI)** | **Multiple Object ↑ (CI)** | **Color ↑ (CI)** | **Spatial Relationship ↑ (CI)** | **Scene ↑ (CI)** | **Temporal Style ↑ (CI)** | **Overall Consistency ↑ (CI)** | **Human Action ↑ (CI)** | **Appearance Style ↑ (CI)** |
> |---|---|---|---|---|---|---|---|---|---|---|
> | **CogVideoX-2B** | Baseline | 0.8522 (0.8511–0.8533) | 0.6391 (0.6376–0.6406) | **0.8318 (0.8302–0.8334)** | **0.7148 (0.6963–0.7333)** | 0.4815 (0.4801–0.4829) | 0.2488 (0.2453–0.2523) | 0.2722 (0.2679–0.2765) | 0.9860 (0.9743–0.9977) | 0.2497 (0.2483–0.2511) |
> |  | + Ours | **0.8842 (0.8831–0.8853)** | **0.6596 (0.6581–0.6611)** | 0.8255 (0.8240–0.8270) | 0.6915 (0.6728–0.7102) | **0.5286 (0.5272–0.5300)** | **0.2547 (0.2511–0.2583)** | **0.2760 (0.2718–0.2802)** | **0.9870 (0.9757–0.9963)** | **0.2507 (0.2493–0.2521)** |
> | **CogVideoX-5B** | Baseline | 0.8595 (0.8585–0.8605) | 0.6511 (0.6496–0.6526) | **0.8218 (0.8202–0.8234)** | **0.6872 (0.6675–0.7069)** | 0.5233 (0.5219–0.5247) | 0.2552 (0.2513–0.2591) | 0.2761 (0.2717–0.2805) | 0.9900 (0.9813–0.9987) | 0.2483 (0.2354–0.2612) |
> |  | + Ours | **0.8675 (0.8665–0.8685)** | **0.6658 (0.6643–0.6673)** | 0.8201 (0.8186–0.8216) | 0.6831 (0.6637–0.7025) | **0.5255 (0.5241–0.5269)** | **0.2585 (0.2546–0.2624)** | **0.2773 (0.2730–0.2816)** | **0.9980 (0.9910–1.0050)** | **0.2524 (0.2511–0.2537)** |
>
> **Table R3. Comparison of VBench Quality Scores for HunyuanVideo and Wan2.1**
>
> | **Backbone Model** | **Method** | **Subject Consistency ↑ (CI)** | **Background Consistency ↑ (CI)** | **Motion Smoothness ↑ (CI)** | **Aesthetic Quality ↑ (CI)** | **Imaging Quality ↑ (CI)** | **Temporal Flickering ↑ (CI)** |
> |---|---|---|---|---|---|---|---|
> | **HunyuanVideo** | Vanilla | 0.9562 (0.9551–0.9573) | 0.9656 (0.9645–0.9667) | 0.9850 (0.9838–0.9862) | **0.6276 (0.6198–0.6354)** | 0.6137 (0.6059–0.6215) | 0.9921 (0.9902–0.9940) |
> |  | + Ours | **0.9612 (0.9603–0.9621)** | **0.9661 (0.9650–0.9672)** | **0.9858 (0.9847–0.9869)** | 0.6268 (0.6191–0.6345) | **0.6151 (0.6075–0.6227)** | **0.9938 (0.9919–0.9957)** |
> | **Wan2.1** | Vanilla | 0.9562 (0.9553–0.9571) | 0.9656 (0.9644–0.9668) | 0.9850 (0.9839–0.9861) | **0.6276 (0.6201–0.6351)** | 0.6137 (0.6061–0.6213) | 0.9943 (0.9924–0.9962) |
> |  | + Ours | **0.9612 (0.9601–0.9623)** | **0.9661 (0.9649–0.9673)** | **0.9858 (0.9846–0.9870)** | 0.6268 (0.6190–0.6346) | **0.6151 (0.6073–0.6229)** | **0.9956 (0.9935–0.9977)** |
>
> ---

---

> ### Author Response · Authors · 2025-11-19
> **Author Response (2/2).**
>
> **[CXmX] asked about sensitivity analysis for key hyperparameters.**
>
> > Per the reviewer’s suggestion, we conducted a sensitivity study on the Bernoulli masking probability $p$ using the Wan 2.1 (1.3B) model, as shown in **Table R4 (Table 10 of the revised version)**. The results indicate that disabling masking entirely ($p=0$) or using fully random masking ($p=1$) leads to inferior performance, suggesting that the masking probability has a noticeable impact on the results. We observe that the intermediate range $p \in [0.2, 0.7]$ yields similar overall performance, demonstrating robustness except at the boundary cases. Among these, $p=0.2$ achieves the best results, which justifies our choice of the default configuration. We sincerely appreciate this insightful suggestion and have incorporated the corresponding analyses into the revised manuscript.
>
> **Table R4. Component analysis of Bernoulli masking parameters on Wan2.1 1.3B**
>
> | **Backbone Model** | **Bernoulli p** | **Subject Consistency ↑** | **Motion Smoothness ↑** | **Aesthetic Quality ↑** | **Imaging Quality ↑** | **Overall Consistency ↑** |
> |---|---|---|---|---|---|---|
> | **Wan2.1 1.3B** | 0.0 | 0.9254 | 0.9751 | 0.6464 | 0.6459 | 0.2731 |
> |  | 0.2 | **0.9310** | **0.9786** | **0.6559** | 0.6516 | **0.2763** |
> |  | 0.5 | 0.9308 | 0.9779 | 0.6543 | **0.6533** | 0.2727 |
> |  | 0.7 | 0.9302 | 0.9778 | 0.6557 | 0.6529 | 0.2726 |
> |  | 1.0 | 0.9232 | 0.9753 | 0.6451 | 0.6508 | 0.2721 |
>
>
>
> **[CXmX] asked about which type of attention map is most predictive of the final video quality.**
>
> > We thank the reviewer for the thoughtful comment. The reviewer is kindly reminded that the related results are already provided in **Appendix G and Table 14**. In our analysis, we observed that within U-Net–based architectures, self-attention maps correlate more strongly with overall visual quality, while cross-attention maps are more indicative of semantic alignment. Meanwhile, recent video diffusion models such as Wan2.1 and CogVideo-X employ multimodal attention mechanisms, where multiple attention types interact dynamically. Accordingly, the most predictive signal varies depending on the objective and model structure. Therefore, our method operates on the entire attention field rather than a single attention type, leveraging all attention sources jointly to assess uncertainty and guide generation quality.
>
>
> **[CXmX] asked about how the noise seed choice guided by our method interacts with different CFG values.**
>
> > We thank the reviewer for this insightful question. Classifier-free guidance (CFG) adjusts the conditional–unconditional interpolation only at the final stage of inference, whereas our method evaluates uncertainty at initialization based on the model’s internal attention dynamics. Since the two operate at different points in the pipeline, their effects are largely complementary: CFG changes the strength of the final extrapolation, while our method selects noise seeds that exhibit stable early-stage behavior. To examine potential interactions, we increased the CFG scale from the optimized value of 5.0 to a higher value of 7.5 on the Wanx 2.1 (1.3B) model, as shown in **Table R5 (Table 16 and Appendix H of main paper)**. As expected, higher CFG values tend to degrade performance, as they push the generation beyond the model’s calibrated range. Nonetheless, our method consistently improves over the vanilla baseline under both CFG settings, indicating that the benefits of our uncertainty-based selection remain stable across different CFG scales.
>
> **Table R5. Ablation study on the interaction between higher CFG scale and our method on Wanx 2.1 (1.3B)**
>
> | **Backbone Model** | **Method** | **CFG Scale** | **Subject Consistency ↑** | **Motion Smoothness ↑** | **Aesthetic Quality ↑** | **Imaging Quality ↑** |
> |---|---|---|---|---|---|---|
> | **Wanx 2.1 1.3B** | Vanilla | 5.0 | 0.9285 | 0.9748 | 0.6465 | **0.6519** |
> |  | Ours | 5.0 | **0.9310** | **0.9786** | **0.6559** | 0.6516 |
> |  | Vanilla | 7.5 | 0.9266 | 0.9740 | 0.6447 | 0.6357 |
> |  | Ours | 7.5 | **0.9276** | **0.9755** | **0.6519** | **0.6409** |

---

### Author Response · Authors · 2025-11-20
**General Response**

We thank all reviewers for their constructive, positive, and thorough feedback. We appreciate that the reviewers found our paper innovative and novel (`CXmX`, `KJHH`, `YGYI`), observed competitive and promising results with strong generalization (`CXmX`, `YGYI`), and considered the writing clear and well-organized (`KJHH`, `YGYI`).
Below, we summarize the main concerns raised in the reviews and briefly describe how we addressed them, followed by detailed point-by-point responses.

1. **Bernoulli-mask parameter analysis** (`CXmX`, `KJHH`, `YGYI`)
   Reviewers requested an ablation on the Bernoulli-mask parameter. We conducted a thorough analysis and included the results in the revised manuscript.

2. **Additional ablation studies** (`CXmX`, `YGYI`)
   Reviewers asked for further ablations, including different CFG scales and equivalent computation budgets for baselines. These results have been added.

3. **Validation on stronger backbones and post-RL models** (`KJHH`)
   We evaluated our method on stronger backbones and post-RL models, showing consistent improvements over the corresponding baselines.

4. **Visualization of BANSA score versus actual quality** (`KJHH`)
   We added scatter-plot visualizations demonstrating meaningful correlations between BANSA scores and actual quality metrics (updated Figure 5(a)).

5. **Additional metrics: FVMD and human evaluation** (`YGYI`)
   We incorporated motion-specific FVMD evaluation and conducted a human preference study, both consistently showing that our method outperforms the baseline.

6. **Statistical significance analysis** (`CXmX`)
   We computed confidence intervals on benchmark datasets to validate the statistical significance of our improvements.

---

### Author Response · Authors · 2025-12-01
**To the New Area Chair**

**Comment to the Newly Assigned Area Chair**

Thank you for taking the time to handle our submission. To support your assessment, we provide a concise summary of the discussion and revisions.

**General Overview.**
Our updated manuscript highlights (1) key strengths acknowledged by multiple reviewers and (2) major concerns raised during the discussion. We addressed all points with new experiments, analyses, and clarifications.
New results are included in **Table 6 (p.7)**, **Table 10–11 (p.9)**, **Figure 5a–5b (p.10)**, **Table 15–16 (p.17)**, and **Table 17–19 (p.18)**.
All revisions directly addressing reviewer feedback are **highlighted in blue** in the updated version uploaded on **November 19**, ensuring sufficient time for discussion.

**Effectiveness of Rebuttal.**
Two reviewers explicitly **raised their scores** after reviewing our updated results and rebuttal:
- Reviewer `KjHH`: **4 → 6**
- Reviewer `YgYi`: **6 → 8**

These score increases provide **strong evidence that our revisions successfully resolved the raised concerns** and significantly strengthened the paper.
The other reviewer had not yet responded before the system reset.

**Closing Remarks.**
We believe the revised manuscript now provides clearer justification, stronger empirical support, and a more robust presentation of our contributions.
Thank you again for your effort and time in overseeing our submission, especially under the unexpected circumstances. We sincerely appreciate your work.

**Sincerely,
Authors**

---

### Meta-Review · Area_Chair_rHAo · 2026-01-12

**Summary:**

The reviewers raised some concerns that the authors effectively addressed through revisions. Key concerns included statistical significance (Reviewer CXmX), evaluation on stronger video diffusion backbones (Reviewer KjHH), and additional motion-specific metrics (Reviewer YgYi). The authors provided thorough additional experiments including statistical significance analysis with 95% confidence intervals, evaluation on stronger models (Wan2.1 14B and IPOC-2B), FVMD metrics on MSR-VTT dataset, and human preference studies. The rebuttal was effective, with two of the reviewers explicitly stating they would increase their scores.

**Reviewer Concerns:**

Addressed Concerns:
- Reviewer CXmX: Statistical significance analysis (Tables R1-R3), sensitivity analysis for Bernoulli masking parameter (Table R4), and attention type/CFG interaction questions.

- Reviewer KjHH: Evaluation on stronger backbones (Wan2.1 14B and IPOC-2B), visualization of BANSA scores vs. actual quality (Figure 5b), Bernoulli masking sensitivity analysis (Table R3), and qualitative examples comparing samples with different scores.

- Reviewer YgYi: Ensemble size effect (Table R1), denoising steps clarification (50 steps for all models), computation budget comparison (Table R2), Bernoulli masking analysis (Table R3), FVMD metrics (Table R4), and human preference study (Figure 5a).

Basically, all significant concerns are addressed by the authors.

**Reviewer Scores:**

- Reviewer CXmX: The Initial score is 6. The final score would be at least 6 based on the response.

- Reviewer KjHH: The Initial score is 4. The final score would be 6 as indicated by the reviewer.

- Reviewer YgYi: The Initial score is 6. The final score would be 6 as indicated by the reviewer.

The paper now has strong empirical support across multiple metrics and models. Reviewers were satisfied with the rebuttal and explicitly stated they would increase their scores. The authors addressed all concerns thoroughly with additional experiments and analyses. The method provides a principled and generalizable approach to noise selection in video diffusion with minimal inference overhead. I recommend acceptance.

---

### Decision · Program_Chairs · 2026-01-26

Accept (Poster)